# Theoretical and Empirical Insights into the Origins of Degree Bias in Graph Neural Networks

**Arjun Subramonian[1], Jian Kang[2], Yizhou Sun[1]**
[1]University of California, Los Angeles, [2]University of Rochester
{`arjunsub, yzsun`}`@cs.ucla.edu`
`jian.kang@rochester.edu`

## Abstract

Graph Neural Networks (GNNs) often perform better for high-degree nodes than low-degree nodes on node classification tasks. This degree bias can reinforce social marginalization by, e.g., privileging celebrities and other high-degree actors in social networks during social and content recommendation. While researchers have proposed numerous hypotheses for why GNN degree bias occurs, we find via a survey of 38 degree bias papers that these hypotheses are often not rigorously validated, and can even be contradictory. Thus, we provide an analysis of the origins of degree bias in message-passing GNNs with different graph filters. We prove that high-degree test nodes tend to have a lower probability of misclassification regardless of how GNNs are trained. Moreover, we show that degree bias arises from a variety of factors that are associated with a node's degree (e.g., homophily of neighbors, diversity of neighbors). Furthermore, we show that during training, some GNNs may adjust their loss on low-degree nodes more slowly than on high-degree nodes; however, with sufficiently many epochs of training, message-passing GNNs can achieve their maximum possible training accuracy, which is not significantly limited by their expressive power. Throughout our analysis, we connect our findings to previously-proposed hypotheses for the origins of degree bias, supporting and unifying some while drawing doubt to others. We validate our theoretical findings on 8 common real-world networks, and based on our theoretical and empirical insights, describe a roadmap to alleviate degree bias. Our code can be found at: `github.com/ArjunSubramonian/degree-bias-exploration`.

## 1 Introduction

Graph neural networks (GNNs) have been applied to node classification tasks such as document topic prediction [4] and content moderation [41]. However, in recent years, researchers have shown that GNNs exhibit better performance for high-degree nodes on node classification tasks. This has significant social implications, such as the marginalization of: (1) authors of less-cited papers when predicting the topic of papers in citation networks; (2) junior researchers when predicting the suitability of prospective collaborators in academic collaboration networks; (3) creators of newer or niche products when predicting the category of products in online product networks; and (4) authors of short or standalone websites when predicting the topic of websites in hyperlink networks.

To illustrate this phenomenon, Figure 1 shows that across different message-passing GNNs (see §D for details about architectures) applied to the CiteSeer dataset (where nodes represent documents and the classification task is to predict their topic), high-degree nodes generally incur a lower test loss than low-degree nodes. In practice, if such GNNs are applied to predict the topic of documents in social scientific studies, less-cited documents will be misclassified, which can lead to the con-

38th Conference on Neural Information Processing Systems (NeurIPS 2024).

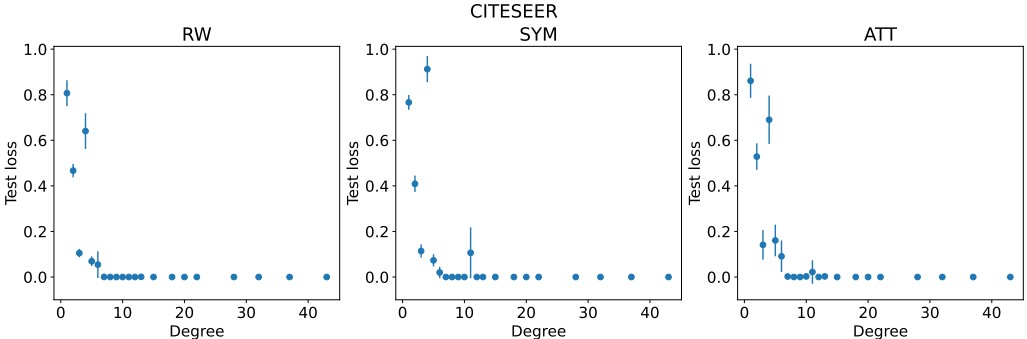

Figure 1: Test loss vs. degree of nodes in CiteSeer for RW, SYM, and ATT GNNs. High-degree nodes generally incur a lower test loss than low-degree nodes do. Error bars are reported over 10 random seeds; all error bars are 1-sigma and represent the standard deviation about the mean.

tributions of their authors not being appropriately recognized and erroneous scientific results. We present additional evidence of degree bias across different GNNs and datasets in §E.

Researchers have proposed various hypotheses for why GNN degree bias occurs in node classification tasks. However, we find via a survey of 38 degree bias papers that these hypotheses are often not rigorously validated, and can even be contradictory (§2). Furthermore, almost no prior works on degree bias provide a comprehensive theoretical analysis of the origins of degree bias that explicitly links a node's degree to its test and training error in the semi-supervised learning setting (§2).

Hence, we theoretically analyze the origins of degree bias in node classification during test and training time for *general* message-passing GNNs, with separate parameters for source and target nodes and residual connections. Our analysis spans different graph filter choices: **RW** (random walk-normalized filter), **SYM** (symmetric-normalized filter), and **ATT** (attention-based filter) (see §D for formal definitions). In particular, we prove that high-degree test nodes tend to have a lower probability of misclassification regardless of how GNNs are trained. Moreover, we show that degree bias arises from a variety of factors that are associated with a node's degree (e.g., homophily of neighbors, diversity of neighbors). Furthermore, we show that during training, SYM (compared to RW) may adjust its loss on low-degree nodes more slowly than on high-degree nodes; however, with sufficiently many epochs of training, message-passing GNNs can achieve their maximum possible training accuracy, which is only trivially curtailed by their expressive power. Throughout our analysis, we connect our findings to previously-proposed hypotheses for the origins of degree bias, supporting and unifying some while drawing doubt to others. We validate our theoretical findings on 8 real-world datasets (see §C)that are commonly used in degree bias papers (see Figure 2, §F). Based on our theoretical and empirical insights, we describe a principled roadmap to alleviate degree bias.

## 2 Background and Related Work

Numerous prior works have proposed hypotheses for why GNN degree bias occurs in node classification tasks. We summarize these hypotheses in Table 2 based on a survey of 38 non-review papers about degree bias in node classification that cite [45], a seminal work on degree bias.

While many of these papers have contributed solutions to degree bias (see §A for a thorough overview), we find that their hypotheses for the origins of degree bias are often not rigorously validated, and can even be contradictory. For example, some hypotheses locate the source of degree bias in the training stage, while others cite interactions between training and test-time factors or purely test-time issues. Moreover, hypothesis (**H5**) in Table 2, which posits that high-degree node representations cluster more strongly, conflicts with and (**H10**), which argues that high-degree node representations have a larger variance. In our theoretical analysis of the origins of degree bias, we connect our findings to these hypotheses.

We further find that almost no prior works on degree bias provide a comprehensive theoretical analysis of the origins of GNN degree bias that explicitly links a node's degree to its test and training

Table 1: Five most popular hypotheses for the origins of degree bias proposed by papers. The remaining hypotheses can be found in Table 2.

| Hypothesis | Papers |
|---|---|
| **(H1)** Neighborhoods of low-degree nodes contain insufficient or overly noisy information for effective representations. | [34], [52], [54], [14], [65], [31], [32], [35], [33], [21], [29], [19], [28], [55], [67], [47], [11], [8], [20], [66], [56] |
| **(H2)** High-degree nodes have a larger influence on GNN training because they have a greater number of links with other nodes, thereby dominating message passing. | [45], [52], [65], [22], [63], [28], [62] |
| **(H3)** High-degree nodes exert more influence on the representations of and predictions for nodes as the number of GNN layers increases. | [65], [6], [27], [11], [64] |
| **(H4)** In semi-supervised learning, if training nodes are picked randomly, test predictions for high-degree nodes are more likely to be influenced by these training nodes because they have a greater number of links with other nodes. | [45], [63], [18] |
| **(H5)** Representations of high-degree nodes cluster more strongly around their corresponding class centers, or are more likely to be linearly separable. | [39], [49], [26] |

error in the semi-supervised learning setting (see Table 3). For example, most works prove that: (a) high-degree nodes have a larger influence on GNN node representations or parameter gradients, or (b) high-degree nodes cluster more strongly around their class centers or are more likely to be linearly separable; however, these works do not directly bound the probability of misclassifying a node during training vs. test time in terms of its degree.

The few works that do provide a theoretical analysis of degree bias: **(A1)** perform this analysis with overly strong assumptions, e.g., that graphs are sampled from a Contextual Stochastic Block Model (CSBM) [10], or **(A2)** posit that GNNs do not have sufficient expressive power to map nodes with different degrees to distinct representations. However, in the case of (1), for CSBM graphs, as the number of nodes $n \to \infty$, the degrees of nodes in each class concentrate around a constant value, which is contradictory to real-world graphs, making CSBM an inappropriate model to theoretically analyze degree bias. Moreover, many real-world social networks exhibit a power-law degree distribution [3], which is not captured by a CSBM. In the case of (2), §I shows that the accuracy of GNNs on real-world networks is not significantly limited by the Weisfeiler-Leman (WL) test, which draws doubt to hypothesis **(H7)**.

Ultimately, previously-proposed hypotheses for why GNN degree bias occurs lack rigorous validation, and can even be contradictory. To unify and distill these hypotheses, we provide an analysis of the origins of degree bias in message-passing GNNs with different graph filters.

## 3   Preliminaries

Throughout our theoretical analysis, we connect our findings to previously-proposed hypotheses for the origins of degree bias, supporting and unifying some while drawing doubt to others. We further validate our findings on 8 real-world datasets (see §C) that are commonly used in degree bias papers (see Figure 2, §F). In all figures (except the PCA plots), error bars are reported over 10 random seeds. The factors of variability include model parameter initialization and training dynamics. All error bars are 1-sigma and represent the standard deviation (not standard error) of the mean. We

implicitly assume that errors are normally-distributed. Error bars are computed using PyTorch's `std` function [40]. We relegate all proofs to §B.

We first introduce relevant notation and assumptions. Suppose we have a $C$-class node classification problem defined over an undirected connected graph $\mathcal{G} = (\mathcal{V}, \mathcal{E})$ with $N = |\mathcal{V}|$ nodes. We assume that our graph structure $\boldsymbol{A} \in \{0, 1\}^{N \times N}$ and node labels $\boldsymbol{Y} \in \mathbb{N}_{\leq C}^{N}$ are fixed, but our node features $\boldsymbol{X} \in \mathbb{R}^{N \times d^{(0)}}$ are independently sampled from class-specific feature distributions, i.e., $\forall i \in \mathcal{V}, \boldsymbol{X}_i \sim \mathcal{D}_{\boldsymbol{Y}_i}$. We further have a model $\mathcal{M}$ that maps $\boldsymbol{X}, \boldsymbol{A}$ to predictions $\widehat{\boldsymbol{Y}} \in \mathbb{R}^{N \times C}$. We use a cross-entropy loss function $\ell(\mathcal{M}|i, c) = -\log \widehat{\boldsymbol{Y}}_{i,c}$ that computes the loss for node $i \in \mathcal{V}$ with respect to class $c$ for $\mathcal{M}$. Per the semi-supervised learning paradigm [24], we train $\mathcal{M}$ with the full graph $\boldsymbol{X}, \boldsymbol{A}$ but only a labeled subset of nodes $S \subset \mathcal{V}$.

## 4 Test-Time Degree Bias

The test-time degree bias of models is important to study, as it can yield disparate performance for low-degree nodes when models are deployed in the real world. We prove that high-degree test nodes tend to have a lower probability of misclassification. Moreover, we show that GNN degree bias arises from a variety of factors that are associated with a node's degree (e.g., homophily of neighbors, diversity of neighbors). We first present a theorem that bounds the probability of a test node $i \in \mathcal{V} \setminus S$ being misclassified. We suppose $\mathcal{M}$ is a neural network that has $L$ layers. It takes as input $\boldsymbol{X}, \boldsymbol{A}$ and generates node representations $\boldsymbol{Z}^{(L)} \in \mathbb{R}^{N \times C}$; these representations are then passed through a softmax activation function to get $\hat{\boldsymbol{Y}} = \boldsymbol{H}^{(L)} = \mathrm{softmax}\left(\boldsymbol{Z}^{(L)}\right)$. At this point, we make few assumptions about the architecture of $\mathcal{M}$; $\mathcal{M}$ could be a graph neural network (GNN), or even an MLP or logistic regression model.

**Theorem 1.** *Consider a test node $i \in \mathcal{V} \setminus S$, with $\boldsymbol{Y}_i = c$. Furthermore, consider a label $c' \neq c$. Let $\mathbb{P}\left(\ell(\mathcal{M}|i, c) > \ell(\mathcal{M}|i, c')\right)$ be the probability of misclassifying $i$. Then, if $\mathbb{E}\left[\boldsymbol{Z}_{i,c'}^{(L)} - \boldsymbol{Z}_{i,c}^{(L)}\right] < 0$ (i.e., $\mathcal{M}$ generalizes in expectation):*

$$\mathbb{P}\left(\ell(\mathcal{M}|i, c) > \ell(\mathcal{M}|i, c')\right) \leq \frac{1}{1 + R_{i,c'}}, \tag{1}$$

*where the squared inverse coefficient of variation $R_{i,c'} = \frac{\left(\mathbb{E}\left[\boldsymbol{Z}_{i,c'}^{(L)} - \boldsymbol{Z}_{i,c}^{(L)}\right]\right)^2}{Var\left[\boldsymbol{Z}_{i,c'}^{(L)} - \boldsymbol{Z}_{i,c}^{(L)}\right]}$.*

The assumption that $\mathcal{M}$ generalizes in expectation is required for the application of Cantelli's inequality in the proof. Notably, it is not possible to prove a similar lower bound without making assumptions about the higher-order moments of $\boldsymbol{Z}_{i,c'}^{(L)} - \boldsymbol{Z}_{i,c}^{(L)}$. The coefficient of variation $\frac{\mathrm{Std}\left[\boldsymbol{Z}_{i,c'}^{(L)} - \boldsymbol{Z}_{i,c}^{(L)}\right]}{\mathbb{E}\left[\boldsymbol{Z}_{i,c'}^{(L)} - \boldsymbol{Z}_{i,c}^{(L)}\right]}$ is a normalized measure of dispersion that is often used in economics to quantify inequality [13]. Thus, $R_{i,c'}$ captures how little $Z_i$ varies relative to its expected value. In summary, the probability of misclassification $\mathbb{P}\left(\ell(\mathcal{M}|i, c) > \ell(\mathcal{M}|i, c')\right)$ can be minimized when $R_{i,c'}$ is maximized. Intuitively, the probability of misclassification is reduced when $\boldsymbol{Z}_i$ is farther away, in expectation, from the decision boundary that separates classes $c$ and $c'$, and has low dispersion. The following subsections reveal why $R_{i,c'}$ is large when $i$ is high-degree.

### 4.1 Random Walk Graph Filter

So far, we have made few assumptions about $\mathcal{M}$. Now, we suppose $\mathcal{M}$ is a general message-passing GNN [16]. In particular, for layer $l$:

$$\boldsymbol{H}^{(l)} = \sigma^{(l)}\left(\boldsymbol{Z}^{(l)}\right) = \sigma^{(l)}\left(\boldsymbol{H}^{(l-1)}\boldsymbol{W}_1^{(l)} + \boldsymbol{P}^{(l)}\boldsymbol{H}^{(l-1)}\boldsymbol{W}_2^{(l)} + \boldsymbol{X}\boldsymbol{W}_3^{(l)}\right), \tag{2}$$

where $\boldsymbol{H}^{(l)} \in \mathbb{R}^{N \times d^{(l)}}$ are the $l$-th layer node representations (with $\boldsymbol{H}^{(0)} = \boldsymbol{X}$ and $d^{(L)} = C$), $\sigma^{(l)}$ is an instance-wise non-linearity (with $\sigma^{(L)}$ being softmax), $\boldsymbol{P}^{(l)} \in \mathbb{R}^{N \times N}$ is a graph filter, and $\boldsymbol{W}_1^{(l)}, \boldsymbol{W}_2^{(l)}, \boldsymbol{W}_3^{(l)} \in \mathbb{R}^{d^{(l-1)} \times d^{(l)}}$ are the $l$-th layer model parameters.

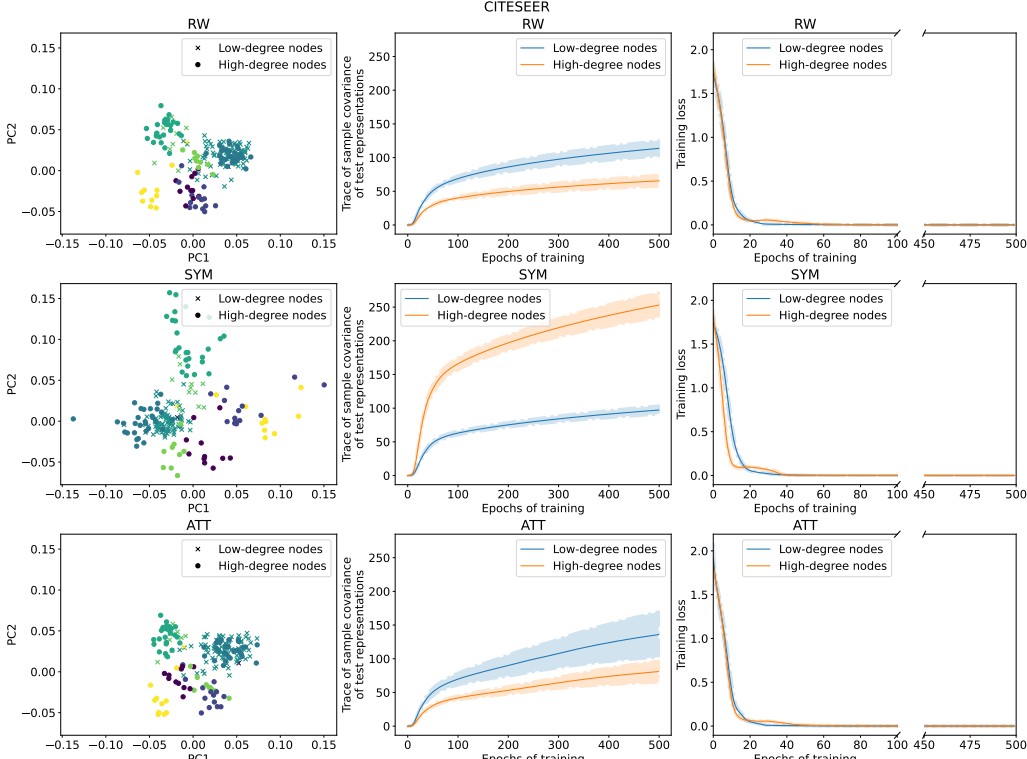

Figure 2: Visual summary of the geometry of representations, variance of representations, and training dynamics of RW, SYM, and ATT GNNs on CiteSeer. We consider low-degree nodes to be the 100 nodes with the smallest degrees and high-degree nodes to be the 100 nodes with the largest degrees. Each point in the plots in the left column corresponds to a test node representation and its color represents the node's class. (In this particular dataset, low-degree nodes are more heavily concentrated in a few classes.) The plots in the left column are based on a single random seed, while the plots in the middle and right columns are based on 10 random seeds. RW representations of low-degree nodes often have a larger variance than high-degree node representations, while SYM representations of low-degree nodes often have a smaller variance. Furthermore, SYM generally adjusts its training loss on low-degree nodes less rapidly.

We first consider the special case that $\forall l \in \mathbb{N}_{\leq L}, \boldsymbol{P}^{(l)} = \boldsymbol{P}_{\text{rw}} = \boldsymbol{D}^{-1}\boldsymbol{A}$ (i.e., the uniform random walk transition matrix), where $\boldsymbol{D}$ is the diagonal degree matrix with entries $\boldsymbol{D}_{ii} = \sum_{j \in \mathcal{V}} \boldsymbol{A}_{ij}$. We further simplify the model by choosing all $\sigma^{(l)}$ ($l < L$) to be the identity function (e.g., as in [51]). By doing so, we get the following linear jumping knowledge model $\overline{\text{RW}}$ [58]:

$$\boldsymbol{H}^{(L)} = \text{softmax}\left(\boldsymbol{Z}^{(L)}\right) = \text{softmax}\left(\sum_{l=0}^{L} \boldsymbol{P}_{\text{rw}}^{l} \boldsymbol{X} \boldsymbol{W}^{(l)}\right), \tag{3}$$

where $\forall l \in \mathbb{N}_{\leq L}, \boldsymbol{W}^{(l)} \in \mathbb{R}^{d^{(0)} \times C}$. $\boldsymbol{W}^{(l)}$ is the sum of all the weight terms that correspond to $\boldsymbol{P}_{rw}^{l}$ in Eqn. 2; for simplicity, we collapse each sum of weight terms into a single weight matrix. It is still reasonable to have a different weight matrix $\boldsymbol{W}^{(l)}$ for each term $\boldsymbol{P}_{rw}^{l} \boldsymbol{X}$, as we may need to extract different information from features aggregated from neighborhoods at different hops. For each model $\mathcal{M}$, $\overline{\mathcal{M}}$ denotes the linearized version of the model that we theoretically analyze. Linearizing GNNs is a common practice in the literature [51, 7, 39].

We now prove a lower bound for $R_{i,c'}$. By identifying nodes for which this lower bound is larger, we can indirectly figure out which nodes have a lower probability of misclassification. In particular, we find that the bound is generally larger for high-degree nodes, which sheds light on the origins of degree bias. For simplicity of notation, we denote the weights corresponding to the decision boundary of the $l$-th term that separates classes $c$ and $c'$ by $\boldsymbol{w}_{c'-c}^{(l)} = \boldsymbol{W}_{.,c'}^{(l)} - \boldsymbol{W}_{.,c}^{(l)}$, and $\mathcal{N}^{(l)}(i)$ to

be the distribution over the terminal nodes of length-$l$ uniform random walks starting from node $i$. We further define:

$$\beta_{i,c'}^{(l)} = \mathbb{E}_{j \sim \mathcal{N}^{(l)}(i)} \left[ \mathbb{E}_{\boldsymbol{x} \sim \mathcal{D}_{\boldsymbol{Y}_j}} \left[ \boldsymbol{x}^T \boldsymbol{w}_{c'-c}^{(l)} \right] \right] \tag{4}$$

as the $l$-hop prediction homogeneity of $i$ with respect to $c'$ when $\boldsymbol{Y}_i = c$. In essence, $\mathbb{E}_{\boldsymbol{x} \sim \mathcal{D}_{\boldsymbol{Y}_j}} \left[ \boldsymbol{x}^T \boldsymbol{w}_{c'-c}^{(l)} \right]$ captures the expected prediction score of $\boldsymbol{w}_{c'-c}^{(l)}$ for a node $j$ whose features $\boldsymbol{X}_j \sim \mathcal{D}_{\boldsymbol{Y}_j}$; when $\mathbb{E}_{\boldsymbol{x} \sim \mathcal{D}_{\boldsymbol{Y}_j}} \left[ \boldsymbol{x}^T \boldsymbol{w}_{c'-c}^{(l)} \right]$ is more negative on average, $\boldsymbol{w}_{c'-c}^{(l)}$ predicts $j$ to belong to class $c$ with higher likelihood. Thus, $\beta_{i,c'}^{(l)}$ measures the expected prediction score for nodes $j$, weighted by their probability of being reached by a length-$l$ random walk starting from $i$.

From a topological perspective, because $\beta_{i,c'}^{(l)}$ depends on the distribution of random walks from $i$, it is intimately related to local graph structure. Indeed, $\beta_{i,c'}^{(l)}$ can be interpreted as a "local subgraph difference" and is highly influenced by the local homophily of $i$. However, $\beta_{i,c'}^{(l)}$ is also influenced by the presence of $l$-hop neighbors contained in the training set, as the model is more likely to correctly classify these nodes by a large margin; hence, $\beta_{i,c'}^{(l)}$ does not *only* boil down to local homophily. We discuss other connections between prediction homogeneity, homophily, and separability in §A.4.

In addition to the $l$-hop prediction homogeneity, we denote the $l$-hop collision probability by:

$$\alpha_i^{(l)} = \sum_{j \in \mathcal{V}} \left[ \left( \boldsymbol{P}_{\text{rw}}^l \right)_{ij} \right]^2, \tag{5}$$

which quantifies the probability of two length-$l$ random walks starting from $i$ colliding at the same end node $j$. When the collision probability is lower, random walks starting from $i$ have a higher likelihood of ending at distinct nodes; in effect, the random walks can be considered to be more diverse.

**Theorem 2.** *Assume that $\forall l \in \mathbb{N}_{\leq L}, \forall j \in \mathcal{V}, Var_{\boldsymbol{x} \sim \mathcal{D}_{\boldsymbol{Y}_j}} \left[ \boldsymbol{x}^T \boldsymbol{w}_{c'-c}^{(l)} \right] \leq M$. Then:*

$$R_{i,c'} \geq \frac{\left( \sum_{l=0}^{L} \beta_{i,c'}^{(l)} \right)^2}{M(L+1) \sum_{l=0}^{L} \alpha_i^{(l)}}. \tag{6}$$

We observe that to make $R_{i,c'}$ larger, and thus minimize the probability of misclassification, it is sufficient (although not necessary) that the inverse collision probability $\frac{1}{\sum_{l=0}^{L} \alpha_i^{(l)}}$ is larger. When $L = 1$, $\frac{1}{\sum_{l=0}^{L} \alpha_i^{(l)}} = \frac{1}{\frac{1}{\mathcal{D}_{ii}}+1}$, which is larger for high-degree nodes. We find empirically that the inverse collision probability is positively associated with node degree (see Figures 3, 13, 14). (We elaborate on connections between the inverse collision probability and node degree in §K.) Furthermore, disparities in the inverse collision probability across nodes with different degrees is *reduced* by residual connections and *increased* by self-loops. Intuitively, random walks starting from high-degree nodes diffuse more quickly, maximizing the probability of any two random walks not colliding at the same end node; in this way, a higher inverse collision probability indicates a more diverse and possibly informative $L$-hop neighborhood. This finding supports hypothesis **(H1)** (see Table 2).

Additionally, to make $R_{i,c'}$ larger, it is sufficient that for all $l \in \mathbb{N}_{\leq L}$, $\beta_{i,c'}^{(l)}$ is more negative, e.g., when most nodes in the $l$-hop neighborhood of $i$ are predicted to belong to class $c$. Thus, $\beta_{i,c'}^{(l)}$ can be more negative when nodes in the $l$-hop neighborhood of $i$ also are in class $c$ (i.e., node $i$ has high local homophily) and were part of the training set $S$, leading to them being correctly classified. This finding supports hypotheses **(H4)** and **(H6)**. Notably, we cannot make $\sum_{l=0}^{L} \beta_{i,c'}^{l}$ more positive to increase $R_{i,c'}$; this would violate the assumption of Theorem 2 that the model generalizes in expectation, which is necessary to make a mathematically rigorous statement about degree bias via tail bounds. Intuitively, it also would not make sense that RW and SYM reduce the misclassification error for a node by predicting its neighbors to be of a different class, since message passing smooths the representations of adjacent nodes. Moreover, distribution shifts in local homophily from train to

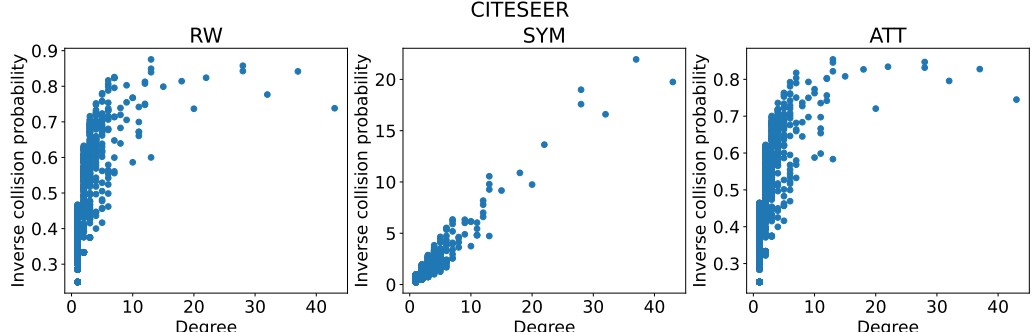

Figure 3: Inverse collision probability vs. degree of nodes in CiteSeer for RW, SYM, and ATT GNNs. Node degrees generally have a strong association with inverse collision probabilities.

test time can reduce test-time prediction performance, bringing $\beta_{i,c'}^{(l)}$ closer to 0; this can increase $R_{i,c'}$, thereby not inducing as much degree bias at the expense of overall test performance.

Furthermore, our proof of Theorem 2 (Eqn. 21) reveals that in expectation, the linearized model $\overline{\text{RW}}$ produces similar representations for low and high-degree nodes with similar $L$-hop neighborhood homophily levels. However, low-degree nodes (specifically nodes with a lower inverse collision probability) tend to have a higher variance in $\overline{\text{RW}}$'s representation space than high-degree nodes do (Eqn. 24). This entails that factors beyond homophily (e.g., diversity of neighbors) induce degree bias. We validate these findings empirically in Figure 2 and §F. In Figure 2, we see in the left plot in the RW row (first row) that low-degree test nodes have representations that are similarly centered but more spread out in the first two principal components of all the test representations than high-degree nodes; we confirm that low-degree node representations have a larger variance in the middle plot in the RW row. Thus, regardless of how RW is trained, low-degree nodes have a higher probability of being on the wrong side of RW's decision boundaries. Indeed, the left plot in the RW row shows that low-degree nodes of a certain class end up closer to nodes of a different class at a higher rate. Notably, this occurs even when RW is relatively shallow (i.e., 3 layers). Thus, this finding supports hypothesis (H5), as well as draws doubt to hypotheses (H3) and (H10). Our results for $\overline{\text{RW}}$ may also hold for ATT when low-degree nodes are generally less attended to since like random walk transition matrices, attention matrices are row-stochastic.

## 4.2 Symmetric Graph Filter

We now consider the special case that $\forall l \in \mathbb{N}_{\leq L}, \boldsymbol{P}^{(l)} = \boldsymbol{P}_{\text{sym}} = \boldsymbol{D}^{-\frac{1}{2}} \boldsymbol{A} \boldsymbol{D}^{-\frac{1}{2}}$. We once again simplify $\mathcal{M}$ by making all $\sigma^{(l)}$ the identity function, getting $\overline{\text{SYM}}$:

$$\boldsymbol{H}^{(L)} = \text{softmax}\left(\boldsymbol{Z}^{(L)}\right) = \text{softmax}\left(\sum_{l=0}^{L} \boldsymbol{P}_{\text{sym}}^l \boldsymbol{X} \boldsymbol{W}^{(l)}\right). \tag{7}$$

We define:

$$\widetilde{\beta}_{i,c'}^{(l)} = \mathbb{E}_{j \sim \mathcal{N}^{(l)}(i)}\left[\frac{1}{\sqrt{\boldsymbol{D}_{jj}}} \mathbb{E}_{\boldsymbol{x} \sim \mathcal{D}_{\boldsymbol{Y}_j}}\left[\boldsymbol{x}^T \boldsymbol{w}_{c'-c}^{(l)}\right]\right] \tag{8}$$

as the degree-discounted $l$-hop prediction homogeneity. Similar to $\beta_{i,c'}^{(l)}$, $\widetilde{\beta}_{i,c'}^{(l)}$ measures the expected prediction score for nodes $j$, but weighted by the inverse square root of their degree in addition to their probability of being reached by a length-$l$ random walk starting from $i$. In effect, $\widetilde{\beta}_{i,c'}^{(l)}$ more heavily discounts the prediction scores for high-degree nodes. We also denote the degree-discounted sum of collision probabilities by:

$$\widetilde{\alpha}_i^{(l)} = \sum_{j \in \mathcal{V}} \frac{1}{\boldsymbol{D}_{jj}}\left[\left(\boldsymbol{P}_{\text{rw}}^l\right)_{ij}\right]^2, \tag{9}$$

where each summation term $\left[\left(\boldsymbol{P}_{\mathrm{rw}}^l\right)_{ij}\right]^2$ quantifies the probability of two length-$l$ random walks starting from $i$ ending at $j$ and is discounted by the degree of $j$. Compared to the random walk setting, the degree-discounted prediction homogeneity and sum of collision probabilities suppress the contributions of high-degree nodes. We now prove a lower bound for $R_{i,c'}$ for $\overline{\mathrm{SYM}}$.

**Theorem 3.** *Assume that* $\forall l \in \mathbb{N}_{\leq L}, \forall j \in \mathcal{V}, Var_{\boldsymbol{x} \sim \mathcal{D}_{\boldsymbol{Y}_j}}\left[\boldsymbol{x}^T \boldsymbol{w}_{c'-c}^{(l)}\right] \leq M$. *Then:*

$$R_{i,c'} \geq \frac{\left(\sum_{l=0}^L \widetilde{\beta}_{i,c'}^{(l)}\right)^2}{M(L+1)\sum_{l=0}^L \widetilde{\alpha}_i^{(l)}}. \tag{10}$$

Once again, we observe that $R_{i,c'}$ is larger, and thus the probability of misclassification is minimized, when the inverse (degree-discounted) sum of collision probabilities $\frac{1}{\sum_{l=0}^L \widetilde{\alpha}_i^{(l)}}$ is larger and for all $l \in \mathbb{N}_{\leq L}$, the (degree-discounted) $l$-hop prediction homogeneity $\widetilde{\beta}_{i,c'}^{(l)}$ is more negative. Like for RW, these findings support hypotheses **(H1)**, **(H4)**, and **(H6)** (see Table 2).

Furthermore, our proof of Theorem 3 (Eqn. 33) reveals that in expectation, $\overline{\mathrm{SYM}}$ often produces representations for low-degree nodes that lie closer to $\overline{\mathrm{SYM}}$'s decision boundary than representations of high-degree nodes with similar $L$-hop neighborhood homophily levels. This is because $\overline{\mathrm{SYM}}$ produces node representations that are approximately scaled by the square root of the node's degree. However, for the same reason, unlike for $\overline{\mathrm{RW}}$, low-degree nodes tend to have a lower variance in $\overline{\mathrm{SYM}}$'s representation space than high-degree nodes do (Eqn. 36); this corroborates the findings of [12]. We validate this empirically in Figure 2 and §F on the homophilic datasets (i.e., all datasets except chameleon and squirrel). In Figure 2, we see in the left plot in the SYM row (second row) that low-degree test nodes (particularly low-degree nodes with many high-degree nodes in their $L$-hop neighborhood) have representations that are closer to SYM's decision boundaries but less spread out in the first two principal components of all the test representations than high-degree nodes; we confirm that low-degree node representations have a smaller or comparable variance in the middle plot in the SYM row. We emphasize that while SYM representations of high-degree nodes have a higher variance, this itself is *not* the cause of degree bias; since the standard deviation *and* expectation of SYM node representations are approximately scaled by the same factor, by Theorem 1, the variance of SYM representations of high-degree nodes does not enlarge $R_{i,c'}$ noticeably more than in the RW case.

Notably, our theoretical findings do extend to heterophilic graphs. In particular, high-degree nodes in heterophilic networks (e.g., chameleon and squirrel) do not have higher negative $L$-hop prediction homogeneity levels due to higher local heterophily (see §F), and hence we do not necessarily observe better test performance for them (see Figure 5). None of our theoretical analysis assumes homophilic networks.

Ultimately, like for RW, low-degree nodes (specifically nodes with a lower inverse collision probability) have a larger probability of being on the wrong side of SYM's decision boundaries (regardless of how SYM is trained). Indeed, low-degree nodes of a certain class end up closer to nodes of a different class at a higher rate. Notably, this occurs even when SYM is relatively shallow (i.e., 3 layers). Thus, this finding supports hypothesis **(H5)**, and draws doubt to hypotheses **(H3)**, **(H7)**, and **(H10)**.

## 5 Training-Time Degree Bias

We show that during training, SYM (compared to RW) may adjust its loss on low-degree nodes more slowly than on high-degree nodes. This finding is important because as GNNs are applied to increasingly large networks, only a few epochs of training may be possible due to limited compute; as such, we must ask: which nodes receive superior utility from limited training? Even though we know the labels for training nodes, GNNs may serve as an efficient lookup mechanism for training nodes in deployed systems; thus, if partially-trained, GNNs can perform poorly for low-degree training nodes. We also empirically demonstrate that despite learning at different rates for low vs. high-degree nodes, message-passing GNNs (even those with static filters) can achieve their maximum possible training accuracy, which is not significantly curtailed by their expressive power.

We first demonstrate that during each step of training of $\overline{\text{SYM}}$ with gradient descent, the loss of low-degree nodes is adjusted more slowly than high-degree nodes. We consider the setting that, for all $l \in \mathbb{N}_{\leq L}$, at each training step $t$:

$$\boldsymbol{W}^{(l)}[t+1] \leftarrow \boldsymbol{W}^{(l)}[t] - \eta \frac{\partial \ell[t]}{\partial \boldsymbol{W}^{(l)}[t]}(B[t]), \tag{11}$$

where $\boldsymbol{W}^{(l)}[t]$ is $\boldsymbol{W}^{(l)}$ at training step $t$, $\eta$ is the learning rate, $\ell[t]$ is the model's loss at $t$, and $B[t] \subseteq S$ (where $S \subseteq \mathcal{V}$ is the labeled subset of nodes) is the batch used at step $t$.

Consider a node $i \in \mathcal{V}$, with $\boldsymbol{Y}_i = c$. We define $\boldsymbol{Z}_i^{(L)}[t]$ to be $\boldsymbol{Z}_i^{(L)}$ at timestep $t$. We begin by proving the following lemma, which states that for any model $\mathcal{M}$, $\ell[t](\mathcal{M}|i,c)$ (for all $t$) is $\lambda$-Lipschitz continuous with respect to $\boldsymbol{Z}_i^{(L)}[t]$.

**Lemma 1.** *For all $t$, $\ell[t](\mathcal{M}|i,c)$ is $\lambda$-Lipschitz continuous with respect to $\boldsymbol{Z}_i^{(L)}[t]$ with constant $\lambda = \sqrt{2}$, that is:*

$$|\ell[t+1](\mathcal{M}|i,c) - \ell[t](\mathcal{M}|i,c)| \leq \left\| \boldsymbol{Z}_i^{(L)}[t+1] - \boldsymbol{Z}_i^{(L)}[t] \right\|_2 \tag{12}$$

Now, we move to the main theorem where we bound the change in loss $i$ after an arbitrary training step $t$ (regardless of batching paradigm) in terms of its degree. We denote the residual of the predictions of $\overline{\text{SYM}}$ at step $t$ by $\epsilon[t] = \boldsymbol{H}^{(L)}[t] - \text{onehot}(\boldsymbol{Y}[t])$, where $\boldsymbol{H}^{(L)}[t]$ and $\text{onehot}(\boldsymbol{Y}[t])$ are the submatrices formed from the rows of $\boldsymbol{H}^{(L)}$ and $\text{onehot}(\boldsymbol{Y})$, respectively, that correspond to the nodes in $B[t]$. Furthermore, we denote $\forall l \in \mathbb{N}_{\leq L}$, the expected similarity of the neighborhoods of $i$ and $B[t]$ by $\widetilde{\chi}_i^{(l)} \in \mathbb{R}^{|B[t]|}$, where for $m \in B[t]$, $\left( \widetilde{\chi}_i^{(l)}[t] \right)_m =$ $\sqrt{\boldsymbol{D}_{mm}} \mathbb{E}_{j \sim \mathcal{N}^{(l)}(i), k \sim \mathcal{N}^{(l)}(m)} \left[ \frac{1}{\sqrt{\boldsymbol{D}_{jj}\boldsymbol{D}_{kk}}} \boldsymbol{X}_j \boldsymbol{X}_k^T \right]$. Specifically, $\left( \widetilde{\chi}_i^{(l)}[t] \right)_m$ captures the degree-discounted expected similarity between the raw features of nodes $j$ and $k$ with respect to the $l$-hop random walk distributions of $i \in \mathcal{V}$ and $m \in B[t]$. Notably, our matrix is *pre*-feature aggregation (e.g., unlike [37]).

**Theorem 4.** *The change in loss for $i$ after an arbitrary training step $t$ obeys:*

$$\left| \ell[t+1](\overline{\text{SYM}}|i,c) - \ell[t](\overline{\text{SYM}}|i,c) \right| \leq \sqrt{\boldsymbol{D}_{ii}} \cdot \sqrt{2}\eta \left\| \epsilon[t] \right\|_F \sum_{l=0}^{L} \left\| \widetilde{\chi}_i^{(l)}[t] \right\|_2. \tag{13}$$

As observed, the change (either increase or decrease) in loss for $i$ after an arbitrary training step has a smaller magnitude if $i$ is low-degree. Thus, when $\ell[t+1](\overline{\text{SYM}}|i,c) < \ell[t](\overline{\text{SYM}}|i,c)$ (e.g., if $i \in B[t]$), the loss for $i$ decreases more slowly when $i$ is low-degree. In effect, because the magnitude of SYM node representations is positively associated with node degree while the magnitude of each gradient descent step is the same across nodes, the representations of low-degree nodes experience a smaller change during each step. We additionally notice that the loss for $i$ changes more slowly when the features of nodes in its $L$-hop neighborhood are not similar to the features in the $L$-hop neighborhoods of the nodes in each training batch (i.e., $\sum_{l=0}^{L} \left\| \widetilde{\chi}_i^{(l)}[t] \right\|_2$ is small). Because the $L$-hop neighborhoods of low-degree nodes tend to be smaller than those of high-degree nodes, their neighborhoods often have less overlap with the neighborhoods of training nodes, which can further constrain the rate at which the loss for $i$ changes. Notably, while node degree highly affects the rate of learning, differences in $\widetilde{\chi}$ across nodes due to factors other than degree are also influential.

We confirm these findings empirically in Figure 2 and §F. For all the datasets, when training SYM, the blue curve (i.e., the loss for low-degree nodes) has a less steep rate of decrease than the orange curve (i.e., the loss for high-degree nodes) as the number of epochs increases. For example, in Figure 2, in the case of RW and ATT, the training loss curves for low and high-degree nodes (including error bars) overlap during the first $\sim 20$ epochs of training. However, for SYM, the loss curve for high-degree nodes descends more rapidly than the curve for low-degree nodes. These findings support hypothesis **(H2)** (c.f. Table 2).

In §H, we demonstrate that during each step of training $\overline{\text{RW}}$ with gradient descent, compared to $\overline{\text{SYM}}$, the loss of low-degree nodes in $S$ is not necessarily adjusted more slowly. Furthermore,

in §I, we empirically show that SYM (despite learning at different rates for low vs. high-degree nodes), RW, and ATT can achieve their maximum possible training accuracy, which is often close to 100%; this indicates that expressive power does not significantly limit the accuracy of these models in practice and draws doubt to hypothesis **(H7)**.

## 6 Principled Roadmap to Address Degree Bias

The primary aim of this work is to explore and explain the origins of GNN degree bias, which lacks a principled understanding. Future research can build on the strong theoretical and empirical foundation laid by this paper to propose alleviation strategies for degree bias. In particular, our findings reveal that any alleviation strategies should target the following theoretically-justified criteria, which we have empirically validated on 8 real-world datasets:

- **Maximizing the inverse collision probability of low-degree nodes (e.g., via edge augmentation for low-degree nodes).** Figure 3 and the plots in Section G show strong positive associations between inverse collision probability and degree for the RW, SYM, and ATT filters, and Figure 1 and the plots in Section E show strong negative associations between degree and test loss for the homophilic datasets. Hence, we validate that a higher inverse collision probability is associated with lower test loss, as our theory predicts.

- **Increasing the $L$-hop prediction homogeneity of low-degree nodes (e.g., by ensuring similar label densities in the neighborhoods of low and high-degree nodes).** The lack of degree bias observed in Figure 5 for chameleon and squirrel (which are heterophilic networks), compared to Figure 1 and the plots in Section E, confirms our theoretical finding that under heterophily, the prediction homogeneity for high-degree nodes is closer to 0, so high-degree nodes do not necessarily experience better performance.

- **Minimizing distributional differences (e.g., differences in expectation, variance) in the representations of low and high-degree nodes.** Figures 2 and 6–10 empirically confirm our theoretical finding that disparities in the expectation and variance of node representations are responsible for performance disparities. Figures 11 and 12 suggest that smaller distributional differences among representations (due to heterophily) can alleviate degree bias.

- **Reducing training discrepancies with regards to the rate at which GNNs learn for low vs. high-degree nodes.** Figure 2 and the plots in Section F validate our theoretical finding that SYM adjusts its loss on low-degree nodes more slowly than on high-degree nodes (see 5).

These criteria are important because they reflect (to a large extent) inherent fairness issues with the graph filters that are popular in graph learning. For instance, the random walk and symmetric filters disadvantage low-degree nodes by yielding representations with high variance and low magnitude, respectively. It is valuable for graph learning practitioners to investigate filters that are adaptive or not restricted to the graph topology in a way that ensures that low-degree nodes are not marginalized through disparate representational distributions or poor neighborhood diversity.

## 7 Conclusion

Our theoretical analysis aims to unify and distill previously-proposed hypotheses for the origins of GNN degree bias. We prove that high-degree test nodes tend to have a lower probability of misclassification and that degree bias arises from a variety of factors associated with a node's degree (e.g., homophily of neighbors, diversity of neighbors). Furthermore, we show that during training, some GNNs may adjust their loss on low-degree nodes more slowly; however, GNNs often achieve their maximum possible training accuracy and are trivially limited by their expressive power. We validate our theoretical findings on 8 real-world networks. Finally, based on our theoretical and empirical insights, we describe a roadmap to alleviate degree bias. More broadly, we encourage research efforts that unveil forms of inequality reinforced by GNNs. We detail the limitations and possible future directions of our work in §L, including our survey, theoretical analysis (e.g., focusing on linearized GNNs, node classification), empirical validation (e.g., exploring degree bias in the inductive learning setting and heterogeneous and directed networks), and roadmap. We additionally discuss broader impacts in §M.

**Acknowledgments and Disclosure of Funding**

This work was partially supported by NSF 2211557, NSF 1937599, NSF 2119643, NSF 2303037, NSF 2312501, NASA, SRC JUMP 2.0 Center, Amazon Research Awards, and Snapchat Gifts.

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

# Appendix

## Table of Contents

# A Overview of Hypotheses for, Theoretical Analyses of, and Proposed Solutions to Degree Bias

## A.1 Hypotheses for Degree Bias

Table 2: Full taxonomy of the hypotheses for the origins of GNN degree bias proposed by papers.

| Hypothesis | Papers |
| --- | --- |
| **(H1)** Neighborhoods of low-degree nodes contain insufficient or overly noisy information for effective representations. | [34], [52], [54], [14], [65], [31], [32], [35], [33], [21], [29], [19], [28], [55], [67], [47], [11], [8], [20], [66], [56] |
| **(H2)** High-degree nodes have a larger influence on GNN training because they have a greater number of links with other nodes, thereby dominating message passing. | [45], [52], [65], [22], [63], [28], [62] |
| **(H3)** High-degree nodes exert more influence on the representations of and predictions for nodes as the number of GNN layers increases. | [65], [6], [27], [11], [64] |
| **(H4)** In semi-supervised learning, if training nodes are picked randomly, test predictions for high-degree nodes are more likely to be influenced by these training nodes because they have a greater number of links with other nodes. | [45], [63], [18] |
| **(H5)** Representations of high-degree nodes cluster more strongly around their corresponding class centers, or are more likely to be linearly separable. | [39], [49], [26] |
| **(H6)** Neighborhoods of high-degree nodes contain more homophilic links, enhancing their representations. | [29], [55] |
| **(H7)** Nodes with different degrees are not necessarily mapped to distinct representations. | [53] |
| **(H8)** Low-degree nodes have class-imbalanced training samples, yielding worse generalization. | [60] |
| **(H9)** High-degree nodes are more likely to be labeled during training and thus GNNs generalize better for them. | [9] |
| **(H10)** Representations of high-degree nodes have higher variance. | [28] |
| **(H11)** Low-degree nodes are more likely to be sampled during training/inference. | [47] |

## A.2 Theoretical Analyses of Degree Bias

Table 3: A taxonomy of GNN degree bias papers based on whether they theoretically analyze the origins of degree bias, explicitly linking a node's degree to its test and training error.

| Explicit theoretical analysis of origins of degree bias? | Papers |
|---|---|
| Yes | [53], [39], [26] |
| No | [45], [34], [52], [54], [14], [65], [49], [31], [22], [60], [61], [9], [5], [32], [28], [43], [35], [33], [21], [29], [18], [6], [19], [55], [50], [27], [67], [47], [8], [11], [20], [62], [66], [56], [64] |

## A.3 Proposed Solutions to Degree Bias

One line of research has produced neighborhood augmentation strategies. For example, [34] perform feature-adaptive neighborhood translation from high-degree nodes to structurally-limited low-degree nodes to enhance their representations. [52] generate multiple views of node neighborhoods (e.g., via node and edge dropping) and learn to maximize the similarity of representations of different views, towards improving the robustness of low-degree node representations. [21] patch the ego-graphs of low-degree nodes by generating virtual neighbors. [55] self-distill graphs and complete the neighborhoods of low-degree nodes with more homophilic links. Other methods include:

- [67] and [66] generate multiple views of node neighborhoods (e.g., via node and edge dropping) and learn to maximize the similarity of representations of different views, towards improving the robustness of low-degree node representations.
- [49] interpolate additional links for low-degree nodes and purify links for high-degree nodes to balance neighborhood information across nodes with different degrees.
- [31] generate more samples in the local neighborhoods of low-degree nodes, to augment the information in these neighborhoods.
- [63] leverage a Transformer architecture and contrastive learning with augmentations (e.g., via node and edge dropping) to bolster the influence of low-degree nodes during training.
- [43] augment the neighborhoods of low-degree nodes in knowledge graphs with synthetic triples.
- [29] augment the local structure of low-degree nodes by adding homophilic edges.
- [47] generate contextually-dependent neighborhoods based on a node's degree.
- [50] contribute a meta-learning strategy to generate additional edges for low-degree nodes.
- [11] introduce dummy nodes connected to all the nodes in the graph to improve message passing to low-degree nodes.
- [20] introduce a learnable graph augmentation strategy to connect low-degree nodes via more within-community edges, as well as an improved self-attention mechanism.

Another line of research has produced algorithms that normalize graph filters or node representations to minimize distributional differences between the representations of nodes with different degrees. [22] propose pre-processing and in-processing approaches that re-normalize the graph filter to be doubly stochastic. [56] calculate node representation statistics via a hybrid strategy and use these statistics to normalize representations. [28] normalize the representations of low-degree and high-degree nodes to have similar distributions.

A different line of research has produced algorithms that separate the learning process for high and low-degree nodes. [53] embed degree-specific weights and hashing functions within the layers of GNNs that guarantee degree-aware node representations. [45] propose a degree-specific GNN layer, to avoid parameter sharing across nodes with different degrees, and a Bayesian teacher network that generates pseudo-labeled neighbors for low-degree nodes to increase their proximity to labels. [60] learn separate expert models for high and low-degree nodes with class and degree-balanced data

subsets, and distill the knowledge of these experts into two student models that are tailored to low and high-degree nodes.

There exist yet other solutions (e.g., based on adversarial learning, attention) that have been produced. [35] learn debiasing functions that distill and complement the GNN encodings of high and low-degree nodes, respectively. [33] leverage meta-learning to learn to learn representations for low-degree nodes in a locality-aware manner. [18] propose a label proximity score, which they find to be more strongly associated with performance than degree, and learn a new graph structure that reduces discrepancies in label proximity scores across nodes. [64] leverage an attention mechanism to enhance focus on low-degree nodes. Other methods include:

- [65] modify the aggregation weights for neighboring node representations according to node degrees.
- [54] learn custom message passing strategies for nodes with different degrees.
- [14] causally determine if low-degree nodes should "trust" messages from their neighbors.
- [9] learn node representations that are invariant to shifts in local neighborhood distributions.
- [32] optimize the graph's adjacency matrix to reduce an upper bound on degree bias with a minimal decrease in overall accuracy.
- [8] locate anchor nodes that through the introduction of links with them can improve the representations of low-degree nodes.
- [27] propose a popularity-weighted aggregator for Graph Convolutional Networks.
- [19] leverage hop-aware attentive aggregation to attend differently to information at different distances.
- [6] estimate the effect of influential high-degree nodes on the representations of low-degree nodes and remove this effect after each graph convolution.
- [62] use adversarial learning to boost the influence of low-degree nodes.

### A.4 Degree Bias from the Perspectives of Homophily and Topology

Prior research has connected degree bias to homophily and graph topology:

- [59] provides a complementary perspective on the possible performance issues of GNNs that arise from degree disparities in graphs (e.g., low-degree nodes induce oversmoothing in homophilic graphs, while high-degree nodes induce oversmoothing in heterophilic networks). Oversmoothing is related to prediction homogeneity $(\sum_{l=0}^{L} \beta_{i,c'}^{(l)})^2$; for homophilic networks, as the number of layers in a GNN increases (i.e., as oversmoothing occurs), $(\sum_{l=0}^{L} \beta_{i,c'}^{(l)})^2$ gets closer to 0 (i.e., does not increase $R_{i,c'}$), thereby not inducing as much degree bias. In contrast, our theoretical analysis demonstrates that degree bias occurs without oversmoothing and is amplified by high local homophily.
- [38] connects node distinguishability to node degree and homophily by analyzing the intra-class vs. inter-class embedding distance. We discuss similar quantities in §4.1 and §4.2. However, with the exception of Section 3.5, [38] considers the CSBM-H model in its theoretical analysis, which has pitfalls (as we discuss in §2). Moreover, unlike our work, [38] does not explicitly link the misclassification error of a node to its degree in a more general data and model setting.
- [48] analyzes the effect of heterophily on GNNs via class separability, which it characterizes through neighborhood distributions and average node degree. Similar to [38], [48] only considers the HSBM model in its theoretical analysis.
- [30] observes that GNN performance is lower for high-degree nodes under heterophily. We likewise observe this in Figure 5 for chameleon and squirrel (which are heterophilic networks). Moreover, our theoretical analysis explains why degree bias is not observed for heterophilic graphs. In §4.2, we explain that high-degree nodes in heterophilic networks do not have lower $l$-hop prediction homogeneity levels due to higher local heterophily; hence, we do not necessarily observe better performance for them compared to low-degree nodes.

# B  Proofs

## B.1  Theorem 1

*Proof.* Misclassification occurs when $\ell(\mathcal{M}|i, c) > \ell(\mathcal{M}|i, c')$.

$$\mathbb{P}\left(\ell(\mathcal{M}|i, c) > \ell(\mathcal{M}|i, c')\right) = \mathbb{P}\left(-\log \boldsymbol{H}_{i,c}^{(L)} > -\log \boldsymbol{H}_{i,c'}^{(L)}\right) \tag{14}$$

$$= \mathbb{P}\left(\boldsymbol{H}_{i,c}^{(L)} < \boldsymbol{H}_{i,c'}^{(L)}\right) \tag{15}$$

$$= \mathbb{P}\left(\boldsymbol{Z}_{i,c'}^{(L)} - \boldsymbol{Z}_{i,c}^{(L)} > 0\right). \tag{16}$$

If $\mathbb{E}\left[\boldsymbol{Z}_{i,c'}^{(L)} - \boldsymbol{Z}_{i,c}^{(L)}\right] < 0$ (i.e., $\mathcal{M}$ generalizes in expectation), by Cantelli's inequality:

$$\mathbb{P}\left(\ell(\mathcal{M}|i, c) > \ell(\mathcal{M}|i, c')\right) = \mathbb{P}\left(\left(\boldsymbol{Z}_{i,c'}^{(L)} - \boldsymbol{Z}_{i,c}^{(L)}\right) - \mathbb{E}\left[\boldsymbol{Z}_{i,c'}^{(L)} - \boldsymbol{Z}_{i,c}^{(L)}\right] > -\mathbb{E}\left[\boldsymbol{Z}_{i,c'}^{(L)} - \boldsymbol{Z}_{i,c}^{(L)}\right]\right)$$

$$\tag{17}$$

$$\leq \frac{1}{1 + \frac{\left(-\mathbb{E}\left[\boldsymbol{Z}_{i,c'}^{(L)} - \boldsymbol{Z}_{i,c}^{(L)}\right]\right)^2}{\mathrm{Var}\left[\boldsymbol{Z}_{i,c'}^{(L)} - \boldsymbol{Z}_{i,c}^{(L)}\right]}}. \tag{18}$$

We use Cantelli's inequality, rather than Chebyshev's inequality, because Cantelli's inequality is sharper for one-sided bounds.

$\square$

## B.2  Theorem 2

*Proof.* Denoting the $l$-th term in the summation $\boldsymbol{T}^{(l)} = \boldsymbol{P}_{\mathrm{rw}}^l \boldsymbol{X} \boldsymbol{W}^{(l)}$, $\boldsymbol{T}_{i,c}^{(l)} = \sum_{j \in \mathcal{V}} \left( \boldsymbol{P}_{\mathrm{rw}}^l \right)_{ij} \boldsymbol{X}_j \boldsymbol{W}_{.,c}^{(l)}$. It follows by the linearity of expectation that:

$$\mathbb{E}\left[ \boldsymbol{T}_{i,c'}^{(l)} - \boldsymbol{T}_{i,c}^{(l)} \right] = \sum_{j \in \mathcal{V}} \left( \boldsymbol{P}_{\mathrm{rw}}^l \right)_{ij} \cdot \mathbb{E}_{\boldsymbol{x} \sim \mathcal{D}_{\boldsymbol{Y}_j}} \left[ \boldsymbol{x}^T \boldsymbol{w}_{c'-c}^{(l)} \right] \tag{19}$$

$$= \mathbb{E}_{j \sim \mathcal{N}^{(l)}(i)} \left[ \mathbb{E}_{\boldsymbol{x} \sim \mathcal{D}_{\boldsymbol{Y}_j}} \left[ \boldsymbol{x}^T \boldsymbol{w}_{c'-c}^{(l)} \right] \right] \tag{20}$$

$$= \beta_{i,c'}^{(l)}. \tag{21}$$

Furthermore, by the linearity of variance:

$$\mathrm{Var}\left[ \boldsymbol{T}_{i,c'}^{(l)} - \boldsymbol{T}_{i,c}^{(l)} \right] = \sum_{j \in \mathcal{V}} \left[ \left( \boldsymbol{P}_{\mathrm{rw}}^l \right)_{ij} \right]^2 \cdot \mathrm{Var}_{\boldsymbol{x} \sim \mathcal{D}_{\boldsymbol{Y}_j}} \left[ \boldsymbol{x}^T \boldsymbol{w}_{c'-c}^{(l)} \right] \tag{22}$$

$$\leq M \sum_{j \in \mathcal{V}} \left[ \left( \boldsymbol{P}_{\mathrm{rw}}^l \right)_{ij} \right]^2 \tag{23}$$

$$= M \alpha_i^{(l)}. \tag{24}$$

Then, once again by the linearity of expectation and variance:

$$\left( \mathbb{E}\left[ \boldsymbol{Z}_{i,c'}^{(l)} - \boldsymbol{Z}_{i,c}^{(l)} \right] \right)^2 = \left( \sum_{l=0}^{L} \beta_{i,c'}^{(l)} \right)^2, \tag{25}$$

$$\mathrm{Var}\left[ \boldsymbol{Z}_{i,c'}^{(l)} - \boldsymbol{Z}_{i,c}^{(l)} \right] \leq M(L+1) \sum_{l=0}^{L} \alpha_i^{(l)}. \tag{26}$$

Consequently:

$$\frac{\left( \mathbb{E}\left[ \boldsymbol{Z}_{i,c'}^{(l)} - \boldsymbol{Z}_{i,c}^{(l)} \right] \right)^2}{\mathrm{Var}\left[ \boldsymbol{Z}_{i,c'}^{(l)} - \boldsymbol{Z}_{i,c}^{(l)} \right]} \geq \frac{\left( \sum_{l=0}^{L} \beta_{i,c'}^{(l)} \right)^2}{M(L+1) \sum_{l=0}^{L} \alpha_i^{(l)}}. \tag{27}$$

$$\square$$

## B.3 Theorem 3

*Proof.* Re-expressing the $l$-th term $\boldsymbol{T}^{(l)} = \boldsymbol{P}_{\mathrm{sym}}^l \boldsymbol{X} \boldsymbol{W}^{(l)}$ in the summation:

$$\boldsymbol{T}_{i,c}^{(l)} = \sum_{j \in \mathcal{V}} \left( \boldsymbol{D}^{-\frac{1}{2}} \boldsymbol{A} \boldsymbol{D}^{-\frac{1}{2}} \right)_{ij}^l \boldsymbol{X}_j \boldsymbol{W}_{\cdot,c}^{(l)} \tag{28}$$

$$= \sum_{j \in \mathcal{V}} \left( \boldsymbol{D}^{-1} \boldsymbol{A} \right)_{ij}^l \cdot \frac{\sqrt{\boldsymbol{D}_{ii}}}{\sqrt{\boldsymbol{D}_{jj}}} \boldsymbol{X}_j \boldsymbol{W}_{\cdot,c}^{(l)} \tag{29}$$

$$= \sqrt{\boldsymbol{D}_{ii}} \sum_{j \in \mathcal{V}} \left( \boldsymbol{P}_{\mathrm{rw}}^l \right)_{ij} \cdot \frac{1}{\sqrt{\boldsymbol{D}_{jj}}} \boldsymbol{X}_j \boldsymbol{W}_{\cdot,c}^{(l)}. \tag{30}$$

It follows by the linearity of expectation that:

$$\mathbb{E}\left[ \boldsymbol{T}_{i,c'}^{(l)} - \boldsymbol{T}_{i,c}^{(l)} \right] = \sqrt{\boldsymbol{D}_{ii}} \sum_{j \in \mathcal{V}} \left( \boldsymbol{P}_{\mathrm{rw}}^l \right)_{ij} \cdot \frac{1}{\sqrt{\boldsymbol{D}_{jj}}} \mathbb{E}_{\boldsymbol{x} \sim \mathcal{D}_{\boldsymbol{Y}_j}} \left[ \boldsymbol{x}^T \boldsymbol{w}_{c'-c}^{(l)} \right] \tag{31}$$

$$= \sqrt{\boldsymbol{D}_{ii}} \mathbb{E}_{j \sim \mathcal{N}^{(l)}(i)} \left[ \frac{1}{\sqrt{\boldsymbol{D}_{jj}}} \mathbb{E}_{\boldsymbol{x} \sim \mathcal{D}_{\boldsymbol{Y}_j}} \left[ \boldsymbol{x}^T \boldsymbol{w}_{c'-c}^{(l)} \right] \right] \tag{32}$$

$$= \sqrt{\boldsymbol{D}_{ii}} \widetilde{\beta}_{i,c'}^{(l)}. \tag{33}$$

Furthermore, by the linearity of variance:

$$\mathrm{Var}\left[ \boldsymbol{T}_{i,c'}^{(l)} - \boldsymbol{T}_{i,c}^{(l)} \right] = \boldsymbol{D}_{ii} \sum_{j \in \mathcal{V}} \left[ \left( \boldsymbol{P}_{\mathrm{rw}}^l \right)_{ij} \right]^2 \cdot \frac{1}{\boldsymbol{D}_{jj}} \mathrm{Var}_{\boldsymbol{x} \sim \mathcal{D}_{\boldsymbol{Y}_j}} \left[ \boldsymbol{x}^T \boldsymbol{w}_{c'-c}^{(l)} \right] \tag{34}$$

$$\leq \boldsymbol{D}_{ii} M \sum_{j \in \mathcal{V}} \frac{1}{\boldsymbol{D}_{jj}} \left[ \left( \boldsymbol{P}_{\mathrm{rw}}^l \right)_{ij} \right]^2 \tag{35}$$

$$= \boldsymbol{D}_{ii} M \widetilde{\alpha}_i^{(l)}. \tag{36}$$

Then, once again, by the linearity of expectation and variance:

$$\left( \mathbb{E}\left[ \boldsymbol{Z}_{i,c'}^{(l)} - \boldsymbol{Z}_{i,c}^{(l)} \right] \right)^2 = \boldsymbol{D}_{ii} \left( \sum_{l=0}^{L} \widetilde{\beta}_{i,c'}^{(l)} \right)^2, \tag{37}$$

$$\mathrm{Var}\left[ \boldsymbol{Z}_{i,c'}^{(l)} - \boldsymbol{Z}_{i,c}^{(l)} \right] \leq \boldsymbol{D}_{ii} M (L+1) \sum_{l=0}^{L} \widetilde{\alpha}_i^{(l)}. \tag{38}$$

Consequently:

$$\frac{\left( \mathbb{E}\left[ \boldsymbol{Z}_{i,c'}^{(l)} - \boldsymbol{Z}_{i,c}^{(l)} \right] \right)^2}{\mathrm{Var}\left[ \boldsymbol{Z}_{i,c'}^{(l)} - \boldsymbol{Z}_{i,c}^{(l)} \right]} \geq \frac{\left( \sum_{l=0}^{L} \widetilde{\beta}_{i,c'}^{(l)} \right)^2}{M(L+1) \sum_{l=0}^{L} \widetilde{\alpha}_i^{(l)}}. \tag{39}$$

$\square$

## B.4   Lemma 1

*Proof.* Define $g(\boldsymbol{Z}_i^{(L)}[t]) = \nabla_{\boldsymbol{Z}_i^{(L)}[t]} \ell[t](\mathcal{M}|i,c)$. For simplicity of notation, let $\boldsymbol{x} = \boldsymbol{Z}_i^{(L)}[t]$. It is sufficient to show that $\|g(\boldsymbol{x})\|_2 \le \lambda$. By simple derivation, $(g(\boldsymbol{x}))_i = -\frac{\sum_{a \neq i} e^{\boldsymbol{x}_a}}{\sum_b e^{\boldsymbol{x}_b}}$, and for $j \neq i$, $(g(\boldsymbol{x}))_j = -\frac{e^{\boldsymbol{x}_j}}{\sum_b e^{\boldsymbol{x}_b}}$. Then, by Hölder's inequality:

$$\|g(\boldsymbol{x})\|_2^2 = \frac{\left(\sum_{a \neq i} e^{\boldsymbol{x}_a}\right)^2 + \sum_{a \neq i} \left(e^{\boldsymbol{x}_a}\right)^2}{\left(\sum_b e^{\boldsymbol{x}_b}\right)^2} \tag{40}$$

$$\le \frac{2\left(\sum_{a \neq i} e^{\boldsymbol{x}_a}\right)^2}{\left(\sum_b e^{\boldsymbol{x}_b}\right)^2} \le 2. \tag{41}$$

Thus, $\lambda = \sqrt{2}$. $\qquad\square$

## B.5 Theorem 4

*Proof.* By the Lipschitz continuity of $\ell[t](\overline{\mathrm{SYM}}|i,c)$ (Lemma 1) and the triangle inequality:

$$\left|\ell[t+1](\overline{\mathrm{SYM}}|i,c) - \ell[t](\overline{\mathrm{SYM}}|i,c)\right| \leq \lambda \left\|\boldsymbol{Z}_i^{(L)}[t+1] - \boldsymbol{Z}_i^{(L)}[t]\right\|_2 \tag{42}$$

$$\leq \lambda \sum_{l=0}^{L} \left\|\left(\boldsymbol{P}_{\mathrm{sym}}^l \boldsymbol{X}\right)_i \left(\boldsymbol{W}^{(l)}[t+1] - \boldsymbol{W}^{(l)}[t]\right)\right\|_2 \tag{43}$$

$$= \lambda\eta \sum_{l=0}^{L} \left\|\left(\boldsymbol{P}_{\mathrm{sym}}^l \boldsymbol{X}\right)_i \frac{\partial \ell[t]}{\partial \boldsymbol{W}^{(l)}[t]}(B[t])\right\|_2. \tag{44}$$

By simple derivation, we see that $\frac{\partial \ell[t]}{\partial \boldsymbol{W}^{(l)}[t]}(B[t]) = \left(\boldsymbol{P}_{\mathrm{sym}}^l[t]\boldsymbol{X}\right)^T \epsilon[t]$, where $\boldsymbol{P}_{\mathrm{sym}}^l[t] \in \mathbb{R}^{|B[t]| \times N}$ is the submatrix formed from the rows of $\boldsymbol{P}_{\mathrm{sym}}^l$ that correspond to the nodes in $B[t]$. Then, by the sub-multiplicativity of the $L_2$ norm:

$$\left|\ell[t+1](\overline{\mathrm{SYM}}|i,c) - \ell[t](\overline{\mathrm{SYM}}|i,c)\right| \leq \lambda\eta \sum_{l=0}^{L} \left\|\left(\boldsymbol{P}_{\mathrm{sym}}^l\right)_i \boldsymbol{X}\boldsymbol{X}^T \left(\boldsymbol{P}_{\mathrm{sym}}^l[t]\right)^T \epsilon[t]\right\|_2 \tag{45}$$

$$\leq \lambda\eta \left\|\epsilon[t]\right\|_F \sum_{l=0}^{L} \left\|\left(\boldsymbol{P}_{\mathrm{sym}}^l\right)_i \boldsymbol{X}\boldsymbol{X}^T \left(\boldsymbol{P}_{\mathrm{sym}}^l[t]\right)^T\right\|_2. \tag{46}$$

Similarly to the proof of Theorem 3:

$$\left(\boldsymbol{P}_{\mathrm{sym}}^l\right)_i \boldsymbol{X}\boldsymbol{X}^T \left(\boldsymbol{P}_{\mathrm{sym}}^l\right)_m^T = \sum_{j\in\mathcal{V}} \left(\boldsymbol{P}_{\mathrm{sym}}^l\right)_{ij} \sum_{k\in\mathcal{V}} \left(\boldsymbol{P}_{\mathrm{sym}}^l\right)_{mk} \left(\boldsymbol{X}\boldsymbol{X}^T\right)_{jk} \tag{47}$$

$$= \sqrt{\boldsymbol{D}_{ii}\boldsymbol{D}_{mm}} \mathbb{E}_{j\sim\mathcal{N}^{(l)}(i),k\sim\mathcal{N}^{(l)}(m)} \left[\frac{1}{\sqrt{\boldsymbol{D}_{jj}\boldsymbol{D}_{kk}}} \left(\boldsymbol{X}\boldsymbol{X}^T\right)_{jk}\right]. \tag{48}$$

Hence, $\left\|\left(\boldsymbol{P}_{\mathrm{sym}}^l\right)_i \boldsymbol{X}\boldsymbol{X}^T \left(\boldsymbol{P}_{\mathrm{sym}}^l[t]\right)^T\right\|_2 = \sqrt{\boldsymbol{D}_{ii}} \cdot \left\|\widetilde{\chi}_i^{(l)}[t]\right\|_2$, and:

$$\left|\ell[t+1](\overline{\mathrm{SYM}}|i,c) - \ell[t](\overline{\mathrm{SYM}}|i,c)\right| \leq \sqrt{\boldsymbol{D}_{ii}} \cdot \lambda\eta \left\|\epsilon[t]\right\|_F \sum_{l=0}^{L} \left\|\widetilde{\chi}_i^{(l)}[t]\right\|_2. \tag{49}$$

$\square$

## B.6 Theorem 5

*Proof.* By the Lipschitz continuity of $\ell[t](\overline{\text{RW}}|i,c)$ (Lemma 1) and the triangle inequality:

$$\left|\ell[t+1](\overline{\text{RW}}|i,c) - \ell[t](\overline{\text{RW}}|i,c)\right| \leq \lambda \left\|\boldsymbol{Z}_i^{(L)}[t+1] - \boldsymbol{Z}_i^{(L)}[t]\right\|_2 \tag{50}$$

$$\leq \lambda \sum_{l=0}^{L} \left\|\left(\boldsymbol{P}_{\text{rw}}^l \boldsymbol{X}\right)_i \left(\boldsymbol{W}^{(l)}[t+1] - \boldsymbol{W}^{(l)}[t]\right)\right\|_2 \tag{51}$$

$$= \lambda\eta \sum_{l=0}^{L} \left\|\left(\boldsymbol{P}_{\text{rw}}^l \boldsymbol{X}\right)_i \frac{\partial\ell[t]}{\partial\boldsymbol{W}^{(l)}[t]}(B[t])\right\|_2. \tag{52}$$

By simple derivation, we see that $\frac{\partial\ell[t]}{\partial\boldsymbol{W}^{(l)}[t]}(B[t]) = \left(\boldsymbol{P}_{\text{rw}}^l[t]\boldsymbol{X}\right)^T \epsilon[t]$, where $\boldsymbol{P}_{\text{rw}}^l[t] \in \mathbb{R}^{|B[t]|\times N}$ is the submatrix formed from the rows of $\boldsymbol{P}_{\text{rw}}^l$ that correspond to the nodes in $B[t]$. Then, by the sub-multiplicativity of the $L_2$ norm:

$$\left|\ell[t+1](\overline{\text{RW}}|i,c) - \ell[t](\overline{\text{RW}}|i,c)\right| \leq \lambda\eta \sum_{l=0}^{L} \left\|\left(\boldsymbol{P}_{\text{rw}}^l\right)_i \boldsymbol{X}\boldsymbol{X}^T \left(\boldsymbol{P}_{\text{rw}}^l[t]\right)^T \epsilon[t]\right\|_2 \tag{53}$$

$$\leq \lambda\eta \left\|\epsilon[t]\right\|_F \sum_{l=0}^{L} \left\|\left(\boldsymbol{P}_{\text{rw}}^l\right)_i \boldsymbol{X}\boldsymbol{X}^T \left(\boldsymbol{P}_{\text{rw}}^l[t]\right)^T\right\|_2. \tag{54}$$

Similarly to the proof of Theorem 2:

$$\left(\boldsymbol{P}_{\text{rw}}^l\right)_i \boldsymbol{X}\boldsymbol{X}^T \left(\boldsymbol{P}_{\text{rw}}^l\right)_m^T = \sum_{j\in\mathcal{V}} \left(\boldsymbol{P}_{\text{rw}}^l\right)_{ij} \sum_{k\in\mathcal{V}} \left(\boldsymbol{P}_{\text{rw}}^l\right)_{mk} \left(\boldsymbol{X}\boldsymbol{X}^T\right)_{jk} \tag{55}$$

$$= \mathbb{E}_{j\sim\mathcal{N}^{(l)}(i),k\sim\mathcal{N}^{(l)}(m)} \left[\left(\boldsymbol{X}\boldsymbol{X}^T\right)_{jk}\right]. \tag{56}$$

Hence, $\left\|\left(\boldsymbol{P}_{\text{rw}}^l\right)_i \boldsymbol{X}\boldsymbol{X}^T \left(\boldsymbol{P}_{\text{rw}}^l[t]\right)^T\right\|_2 = \left\|\chi_i^{(l)}[t]\right\|_2$, and:

$$\left|\ell[t+1](\overline{\text{RW}}|i,c) - \ell[t](\overline{\text{RW}}|i,c)\right| \leq \lambda\eta \left\|\epsilon[t]\right\|_F \sum_{l=0}^{L} \left\|\chi_i^{(l)}[t]\right\|_2. \tag{57}$$

$\square$

# C  Datasets

In our experiments, we use 8 real-world network datasets from [4], [42], and [41], covering diverse domains (e.g., citation networks, collaboration networks, online product networks, Wikipedia networks). We provide a description and statistics of each dataset in Table 4. All the datasets have node features and are undirected. For each node, we normalize its features to sum to 1, following [15][1]. We were unable to find the exact class names and their label correspondence from the dataset documentation.

- In all the citation network datasets, nodes represent documents, edges represent citation links, and features are a binary bag-of-words representation of documents. The classification task is to predict the topic of documents.

- In the collaboration network datasets, nodes represent authors, edges represent coauthorships, and features are a binary bag-of-word representation of keywords from the authors' papers. The classification task is to predict the most active field of study for authors.

- In the online product network datasets, nodes represent products, edges represent that two products are often purchased together, and features are a binary bag-of-word representation of product reviews. The classification task is to predict the category of products.

- In the Wikipedia network datasets, nodes represent Wikipedia websites, edges represent hyperlinks between them, and features are a binary bag-of-word representation of informative nouns from the pages. The classification task is to predict the level of average daily traffic for pages.

We use PyTorch and PyTorch Geometric to load and process all datasets [40, 15]. Our usage of these libraries and datasets complies with their license. PyTorch and PyTorch Geometric are available under a torch-specific license[2] and MIT license[3], respectively. Cora_ML and CiteSeer are available under an MIT License[4] and can be found here: `https://github.com/abojchevski/graph2gauss/tree/master/data`. CS, Physics, Amazon Photo, and Amazon Computers are available under an MIT License[5] and can be found here: `https://github.com/shchur/gnn-benchmark/tree/master/data/npz`. chameleon and squirrel are available under a GLP-3.0 license[6] and can be found here: `https://graphmining.ai/datasets/ptg/wiki`.

While these datasets are widely used, we did not obtain explicit consent from any data subjects whose data the datasets may contain. To the best of our knowledge (via manual sampling and inspection), the datasets do not contain any personally identifiable information or offensive content.

Table 4: Summary of the datasets used in our experiments.

| Name | Domain | # Nodes | # Edges | # Features | # Classes |
|---|---|---|---|---|---|
| Cora_ML | citation | 2995 | 16316 | 2879 | 7 |
| CiteSeer | citation | 4230 | 10674 | 602 | 6 |
| CS | collaboration | 18333 | 163788 | 6805 | 15 |
| Physics | collaboration | 34493 | 495924 | 8415 | 5 |
| Amazon Photo | online product | 7650 | 238162 | 745 | 8 |
| Amazon Computers | online product | 13752 | 491722 | 767 | 10 |
| chameleon | Wikipedia | 2277 | 36101 | 2325 | 5 |
| squirrel | Wikipedia | 5201 | 217073 | 2089 | 5 |

---

[1]`https://github.com/pyg-team/pytorch_geometric/blob/master/examples/link_pred.py`
[2]`https://github.com/pytorch/pytorch/blob/main/LICENSE`
[3]`https://github.com/pyg-team/pytorch_geometric/blob/master/LICENSE`
[4]`https://github.com/abojchevski/graph2gauss/blob/master/LICENSE`
[5]`https://github.com/shchur/gnn-benchmark/blob/master/LICENSE`
[6]`https://github.com/benedekrozemberczki/MUSAE/blob/master/LICENSE`

# D Models

In our experiments, we transductively learn and compute node representations using encoders based on Graph Convolutional Networks (GCNs) [24], GraphSAGE [17], and Graph Attention Networks (GATs) [46].

In all cases, we use general message-passing GNNs $\mathcal{M}$ [16], which include separate parameters for source and target nodes and residual connections; in particular, for layer $l$:

$$\boldsymbol{H}^{(l)} = \sigma^{(l)}\left(\boldsymbol{Z}^{(l)}\right) = \sigma^{(l)}\left(\boldsymbol{H}^{(l-1)}\boldsymbol{W}_1^{(l)} + \boldsymbol{P}^{(l)}\boldsymbol{H}^{(l-1)}\boldsymbol{W}_2^{(l)} + \boldsymbol{X}\boldsymbol{W}_3^{(l)}\right), \qquad (58)$$

where $\boldsymbol{H}^{(l)} \in \mathbb{R}^{N \times d^{(l)}}$ are the $l$-th layer node representations (with $\boldsymbol{H}^{(0)} = \boldsymbol{X}$ and $d^{(L)} = C$), $\sigma^{(l)}$ is an instance-wise non-linearity (with $\sigma^{(L)}$ being softmax), $\boldsymbol{P}^{(l)} \in \mathbb{R}^{N \times N}$ is a graph filter, and $\boldsymbol{W}_1^{(l)}, \boldsymbol{W}_2^{(l)}, \boldsymbol{W}_3^{(l)} \in \mathbb{R}^{d^{(l-1)} \times d^{(l)}}$ are the $l$-th layer model parameters.

We consider the following special cases of $\mathcal{M}$ which vary with respect to their graph filter:

- **RW:** $\forall l \in \mathbb{N}_{\leq L}, \boldsymbol{P}^{(l)} = \boldsymbol{P}_{\mathrm{rw}} = \boldsymbol{D}^{-1}\boldsymbol{A}$ (i.e., the uniform random walk transition matrix), where $\boldsymbol{D}$ is the diagonal degree matrix with entries $\boldsymbol{D}_{ii} = \sum_{j \in \mathcal{V}} \boldsymbol{A}_{ij}$.

- **SYM:** $\forall l \in \mathbb{N}_{\leq L}, \boldsymbol{P}^{(l)} = \boldsymbol{P}_{\mathrm{sym}} = \boldsymbol{D}^{-\frac{1}{2}}\boldsymbol{A}\boldsymbol{D}^{-\frac{1}{2}}$.

- **ATT:** $\forall l \in \mathbb{N}_{\leq L}, \boldsymbol{P}^{(l)}$ is a graph attentional operator with default hyperparameters and a single head [46].

Each encoder has three layers (64-dimensional hidden layers), with a ReLU nonlinearity in between layers. We do not use any regularization (e.g., Dropout, BatchNorm). The encoders are explicitly trained for node classification with the cross-entropy loss and the Adam optimizer [23] with full-batch gradient descent on the training set. We use a learning rate of 5e-3. We further use a random node split of 1000-500-rest for test-val-train. We train all encoders until they reach the training accuracy of $\mathrm{MAJ}_{\mathrm{WL}}$ and select the model parameters with the highest validation accuracy. Although we do not do any hyperparameter tuning, the test accuracy values indicate that the encoders are well-trained.

We use PyTorch [40] and PyTorch Geometric [15][7] to train all the encoders on a single NVIDIA GeForce GTX Titan Xp Graphic Card with 12196MiB of space on an internal cluster. On average (with respect to the datasets), the median time per training epoch was 0.05 seconds. Thus, the estimated total compute is:

$$
\begin{aligned}
&0.05 \text{ (approximate number of seconds per epoch)} && (59) \\
&\times\, 500 \text{ (number of epochs per training run)} && (60) \\
&\times\, 10 \text{ (number of training runs/random seeds per model)} && (61) \\
&\times\, 3 \text{ (number of models per dataset)} && (62) \\
&\times\, 8 \text{ (number of datasets)} && (63) \\
&\times\, 12196 \text{ MiB (GPU memory)} && (64) \\
&=\, 73,176\text{k MiB} \times \text{seconds} && (65)
\end{aligned}
$$

These experiments were run a few times (e.g., due to the discovery of bugs), but the full research project did not involve other experiments not reported in the paper. We used a single CPU worker to load datasets and plot results. The datasets take up 0.2975 MB of disk space.

---

[7] Our usage of all libraries complies with their license. PyTorch and PyTorch Geometric are available under a torch-specific license[8] and MIT license[9], respectively.

# E Additional Degree Bias Plots

Unlike the other datasets, we do not observe degree bias for chameleon and squirrel because these datasets are heterophilic. We intentionally include these datasets to draw contrast to the other, homophilic datasets and validate our theory. For example, in §4.2, we explain that high-degree nodes in heterophilic networks do not have more negative $l$-hop prediction homogeneity levels due to higher local heterophily levels; hence, we do not necessarily observe better performance for them compared to low-degree nodes.

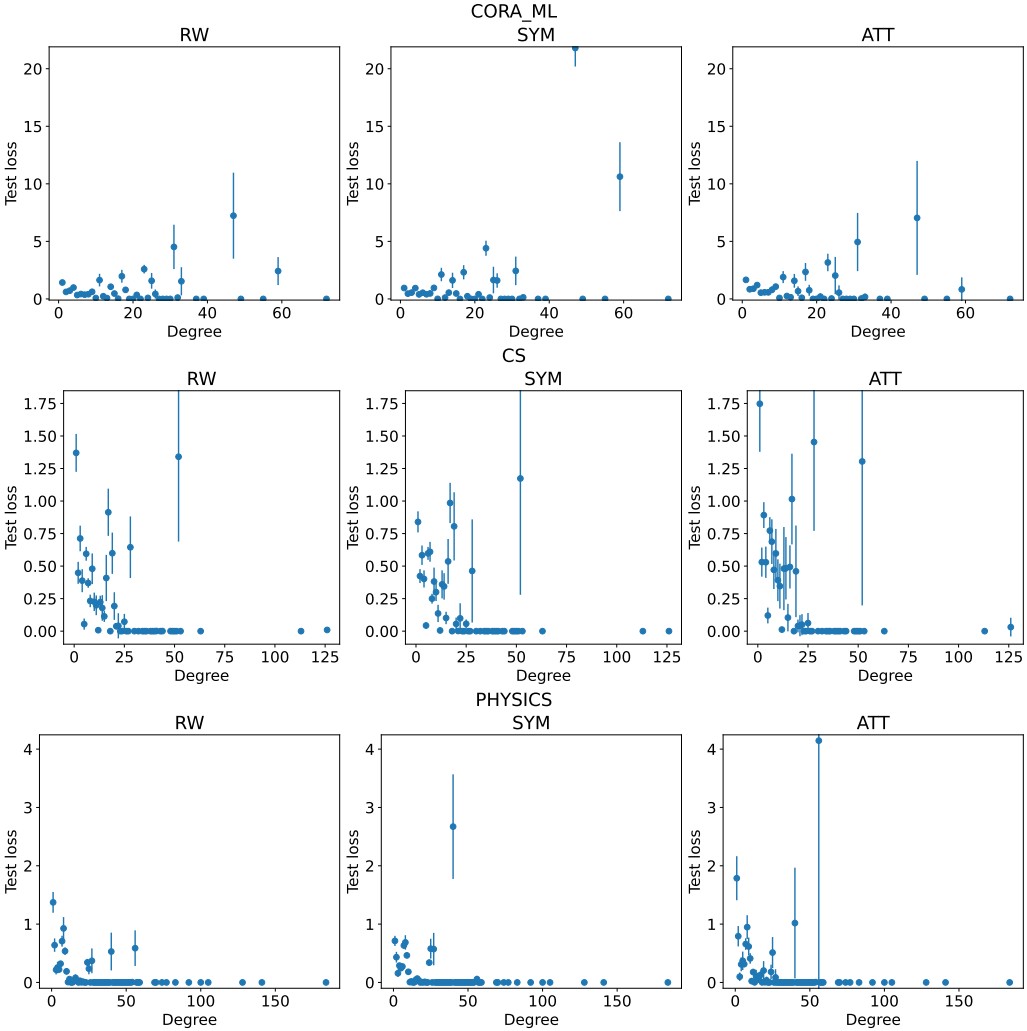

Figure 4: Test loss vs. degree of nodes in citation and collaboration network datasets for RW, SYM, and ATT GNNs. High-degree nodes generally incur a lower test loss than low-degree nodes do. Error bars are reported over 10 random seeds; all error bars are 1-sigma and represent the standard deviation about the mean.

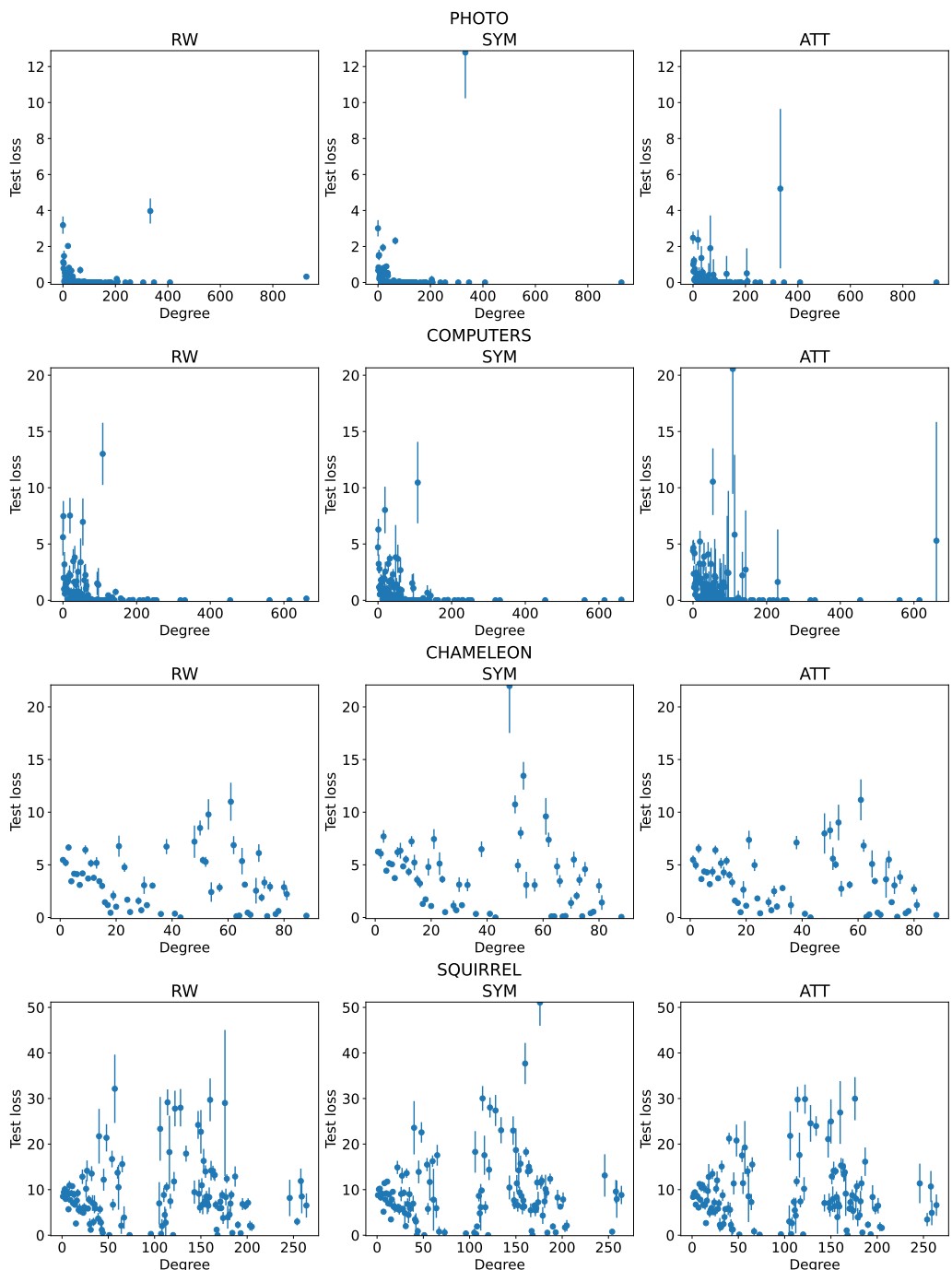

Figure 5: Test loss vs. degree of nodes in online product and Wikipedia network datasets for RW, SYM, and ATT GNNs. High-degree nodes generally incur a lower test loss than low-degree nodes do. Error bars are reported over 10 random seeds; all error bars are 1-sigma and represent the standard deviation about the mean.

# F   Additional Visual Summaries of Theoretical Results

In the plots below, we consider low-degree nodes to be the 100 nodes with the smallest degrees and high-degree nodes to be the 100 nodes with the largest degrees. Each point in the plots in the left column corresponds to a test node representation and its color represents the node's class. The plots in the left column are based on a single random seed, while the plots in the middle and right columns are based on 10 random seeds. RW representations of low-degree nodes often have a larger variance than high-degree node representations, while SYM representations of low-degree nodes often have a smaller variance. Furthermore, SYM generally adjusts its training loss on low-degree nodes less rapidly.

The training loss curves in Figure 6 still support our theoretical analysis. Theorem 4 reveals that for $\overline{SYM}$, node degree *and* the (degree-discounted) expected feature similarity $\widetilde{\chi}_i$ affects the rate of learning. On the other hand, Theorem 5 indicates that for $\overline{RW}$, while we do not expect node degree to impact the rate of learning, the expected feature similarity $\chi_i$ is still influential. Hence, interpreting Theorems 4 and 5 jointly, we expect and accordingly observe that the orange curve for SYM has a steeper rate of decrease *relative* to the orange curve for RW as the number of epochs increases.

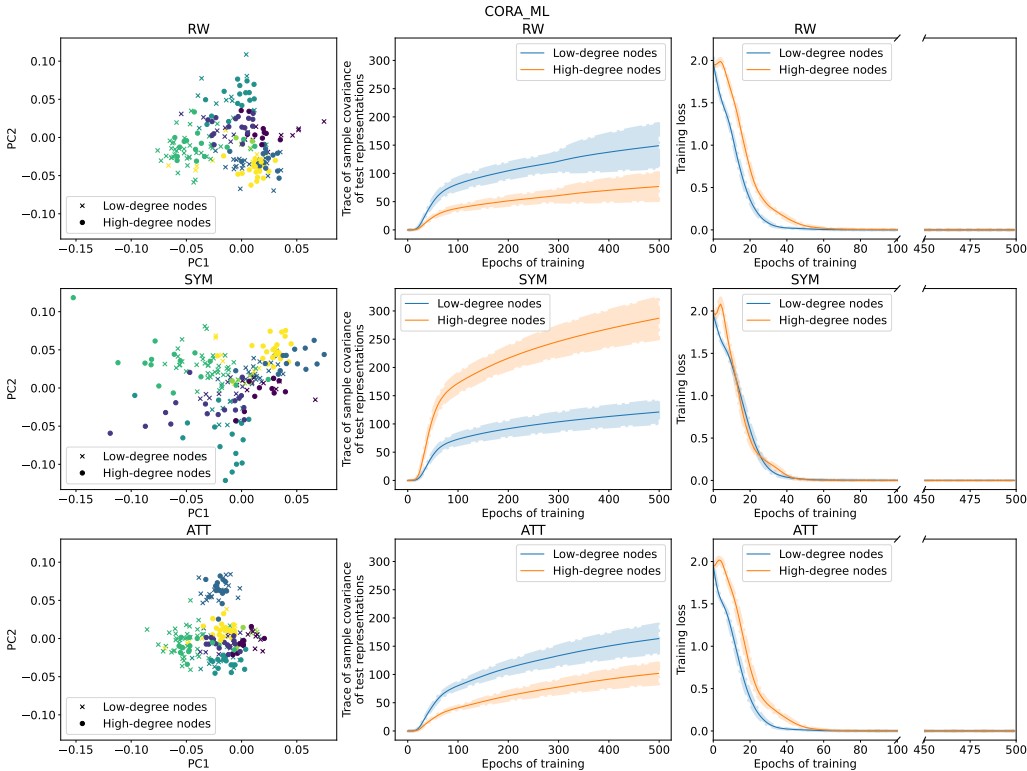

Figure 6: Visual summary of the geometry of representations, variance of representations, and training dynamics of RW, SYM, and ATT GNNs on Cora_ML.

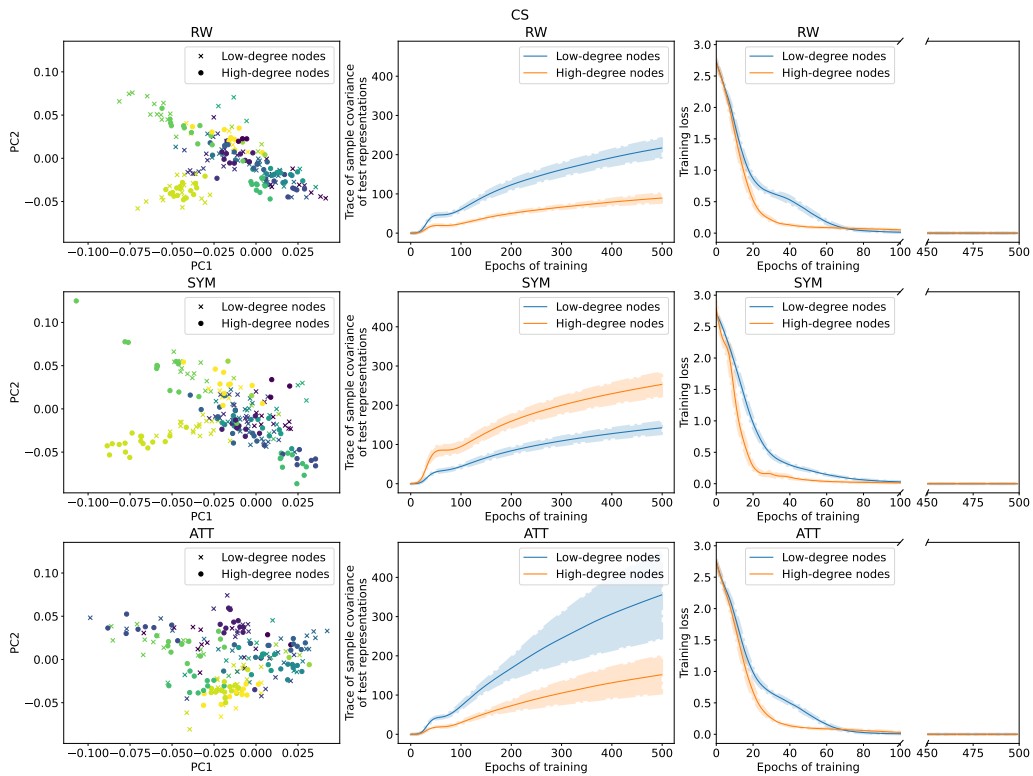

Figure 7: Visual summary of the geometry of representations, variance of representations, and training dynamics of RW, SYM, and ATT GNNs on CS.

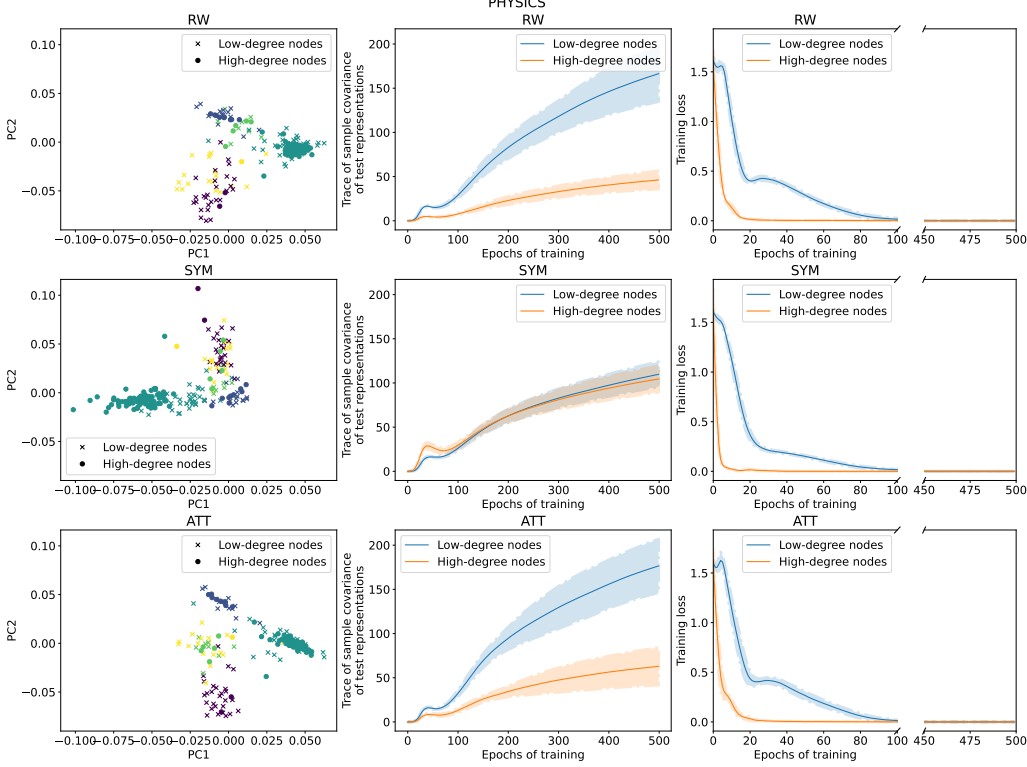

Figure 8: Visual summary of the geometry of representations, variance of representations, and training dynamics of RW, SYM, and ATT GNNs on Physics.

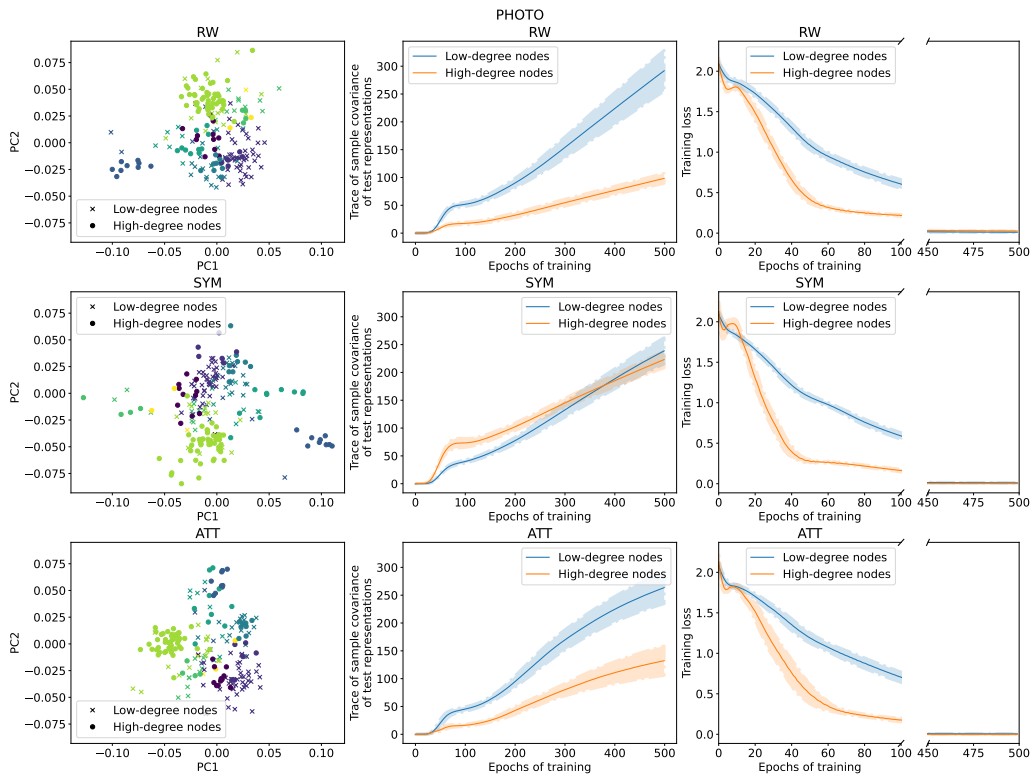

Figure 9: Visual summary of the geometry of representations, variance of representations, and training dynamics of RW, SYM, and ATT GNNs on Amazon Photo.

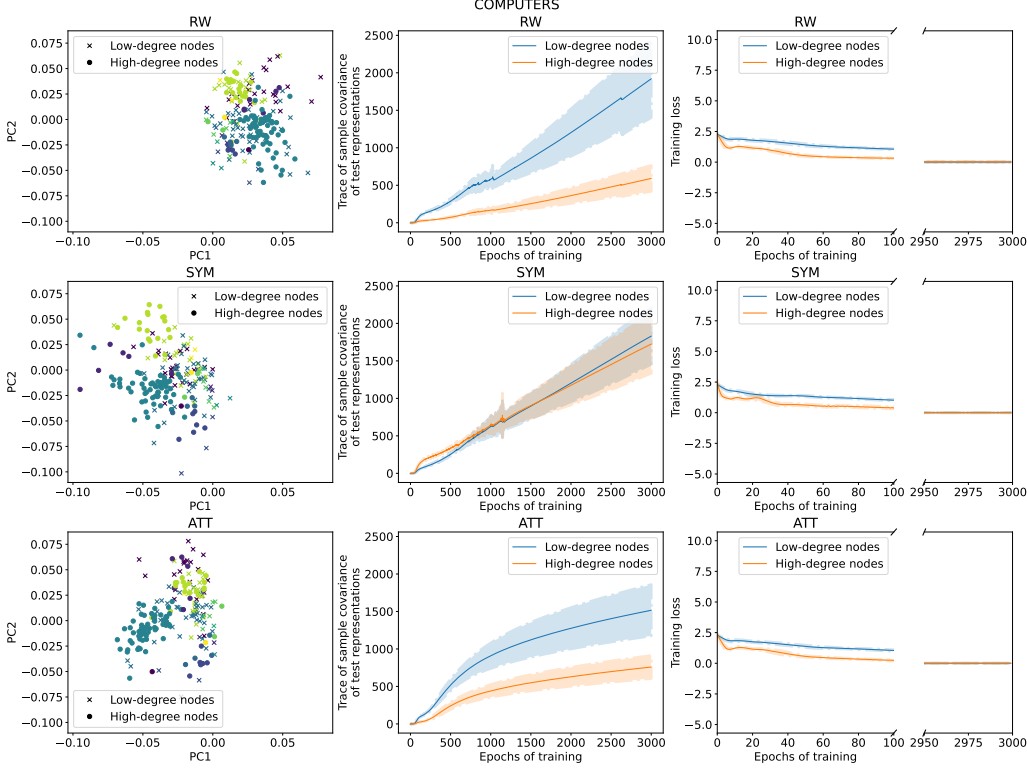

Figure 10: Visual summary of the geometry of representations, variance of representations, and training dynamics of RW, SYM, and ATT GNNs on Amazon Computers.

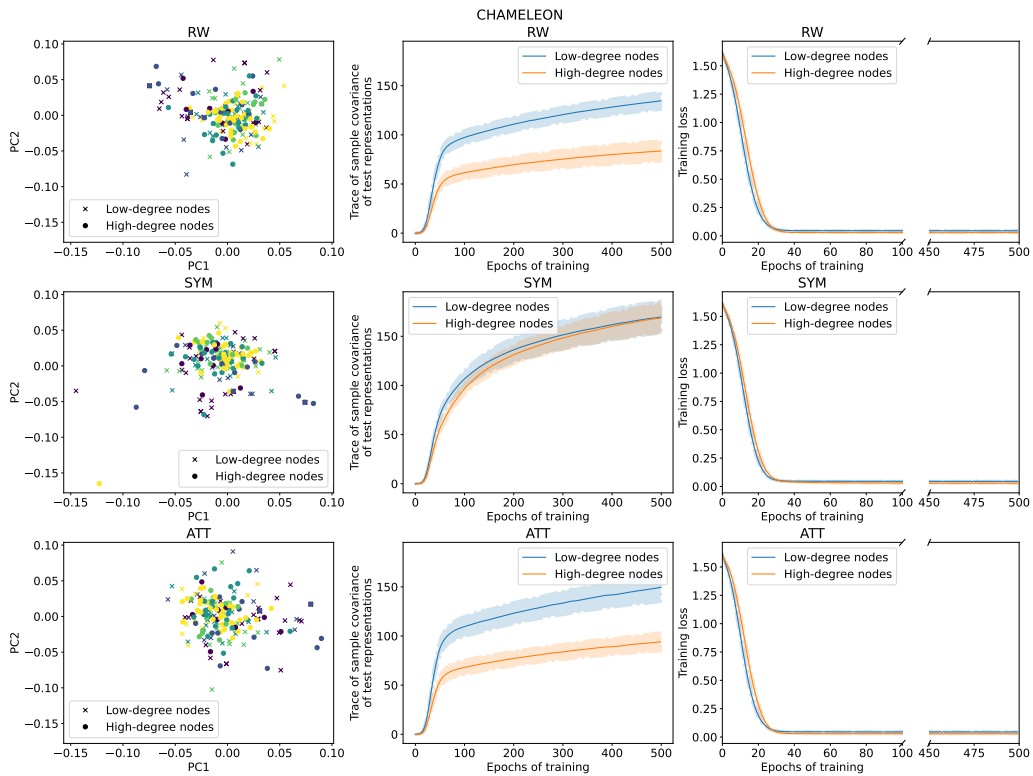

Figure 11: Visual summary of the geometry of representations, variance of representations, and training dynamics of RW, SYM, and ATT GNNs on chameleon.

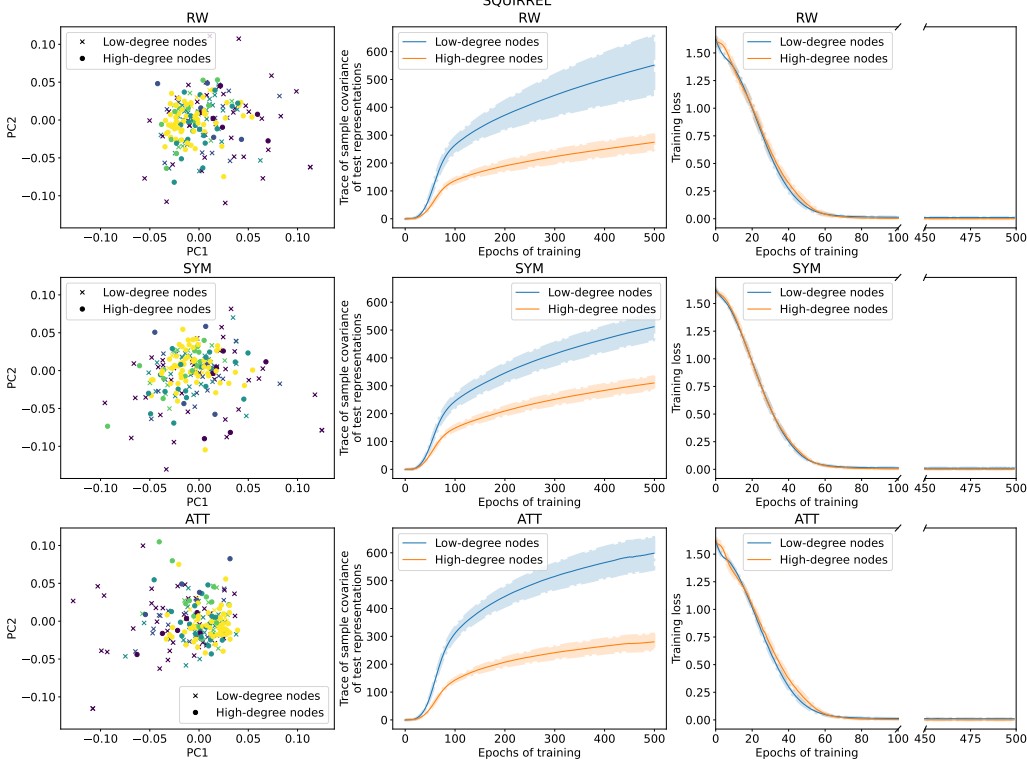

Figure 12: Visual summary of the geometry of representations, variance of representations, and training dynamics of RW, SYM, and ATT GNNs on chameleon.

## G    Additional Inverse Collision Probability Plots

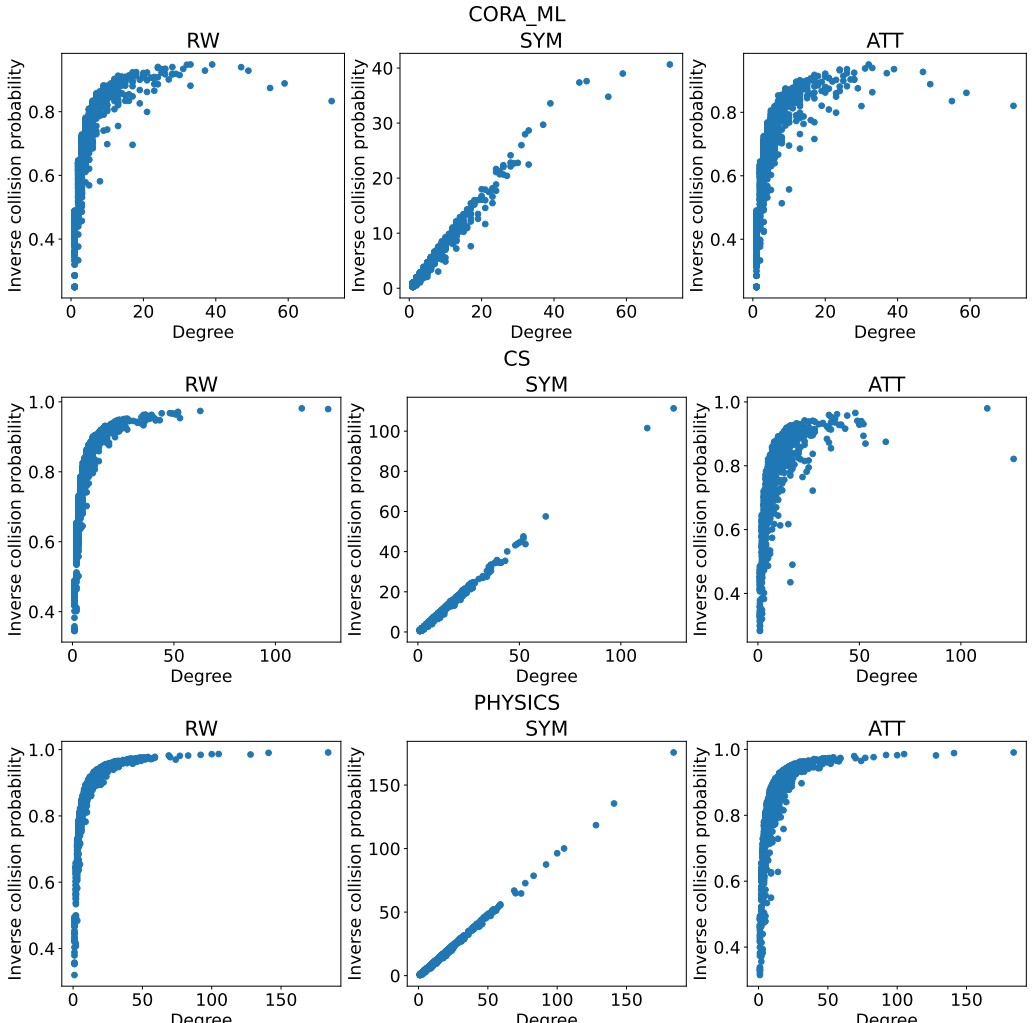

Figure 13: Inverse collision probability vs. degree of nodes in citation and collaboration network datasets for RW, SYM, and ATT GNNs. Node degrees generally have a strong association with inverse collision probabilities.

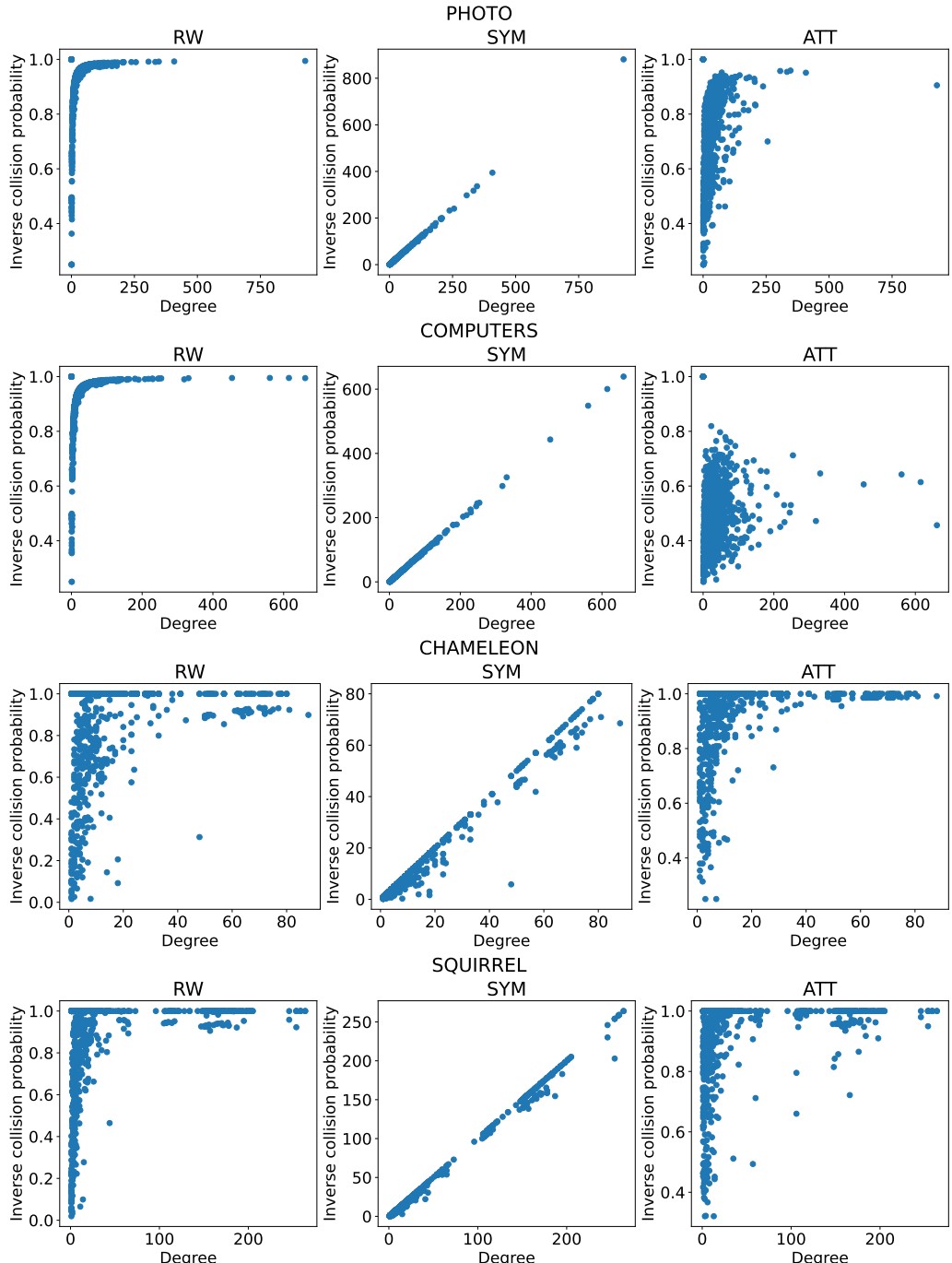

Figure 14: Inverse collision probability vs. degree of nodes in citation and collaboration network datasets for RW, SYM, and ATT GNNs. Node degrees generally have a strong association with inverse collision probabilities.

## H Training-Time Degree Bias: Random Walk Graph Filter

We now demonstrate that during each step of training $\overline{RW}$ with gradient descent, compared to $\overline{SYM}$, the loss of low-degree nodes in $S$ is not necessarily adjusted more slowly. We define $\forall l \in \mathbb{N}_{\leq L}, \chi_i^{(l)} \in \mathbb{R}^{|B[t]|}$, where for $m \in B[t], \left(\chi_i^{(l)}[t]\right)_m = \mathbb{E}_{j \sim \mathcal{N}^{(l)}(i), k \sim \mathcal{N}^{(l)}(m)} \left[X_j X_k^T\right]$. In effect, $\left(\chi_i^{(l)}[t]\right)_m$ captures the expected similarity between the raw features of nodes $j$ and $k$ with respect to the $l$-hop random walk distributions of $i \in \mathcal{V}$ and $m \in B[t]$.

**Theorem 5.** *The change in loss for $i$ after an arbitrary training step $t$ obeys:*

$$\left|\ell[t+1](\overline{RW}|i,c) - \ell[t](\overline{RW}|i,c)\right| \leq \sqrt{2}\eta \|\epsilon[t]\|_F \sum_{l=0}^{L} \left\|\chi_i^{(l)}[t]\right\|_2. \tag{66}$$

For $\overline{RW}$, the change (either increase or decrease) in loss for $i$ after an arbitrary training step does not necessarily have a smaller magnitude if $i$ is low-degree. However, the $L$-hop neighborhoods of low-degree nodes still often have less overlap with the neighborhoods of training nodes, which can constrain the rate at which the loss for $i$ changes. We confirm these findings empirically in Figure 2 and §F. For all the datasets, the blue curve for RW has a less steep rate of decrease than the blue curve for SYM as the number of epochs increases. However, for RW itself, the orange curve generally descends more quickly than the blue curve, with the exception of the heterophilic chameleon and squirrel datasets, for which the features of nodes in the neighborhoods of high-degree nodes and training nodes are dissimilar. Therefore, models do not learn more rapidly for high-degree nodes under heterophily. These findings support hypothesis **(H2)**. Our results for $\overline{RW}$ may also apply to ATT when low-degree nodes are generally attended to less.

# I  Achieving Maximum Training Accuracy

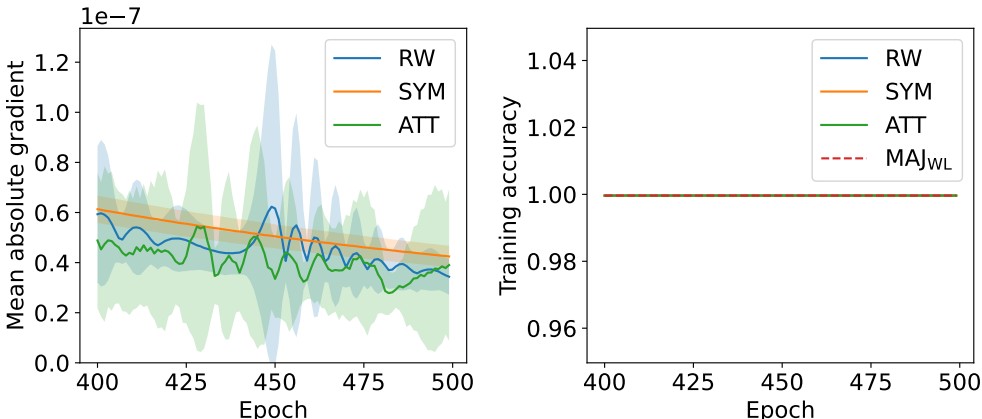

Figure 15: Mean absolute parameter gradient vs. training epoch for RW, SYM, and ATT GNNs on CiteSeer (over 10 random seeds). The training accuracy of SYM, RW, and ATT ultimately reach the accuracy of $\text{MAJ}_{\text{WL}}$.

We now empirically show that SYM (despite learning at different rates for low vs. high-degree nodes), RW, and ATT can achieve their maximum possible training accuracy (i.e., the accuracy of a majority voting-classifier $\text{MAJ}_{\text{WL}}$). Furthermore, per our experiments, the accuracy of $\text{MAJ}_{\text{WL}}$ is often close to 100%, indicating that the WL test does not significantly limit the accuracy of SYM, RW, and ATT in practice.

Per [57], because the expressive power of SYM, RW, and ATT are limited by the WL test, an upper bound on their training accuracy is the accuracy of a majority voting-classifier $\text{MAJ}_{\text{WL}}$ applied to WL node colorings (for details on how to compute colorings, see §II.A and §VI.B of [68]). In particular, if the WL test produces the colors $\boldsymbol{c} \in \mathbb{K}^{|S|}$ for nodes in $S$, $\text{MAJ}_{\text{WL}}$ predicts node $i$ to have the label $\widehat{\boldsymbol{Y}}_i = \text{MAJ}_{\text{WL}}(i, \boldsymbol{X}, \boldsymbol{A}) = \text{mode}\{\boldsymbol{Y}_j | j \in \mathcal{V}, \boldsymbol{c}_j = \boldsymbol{c}_i\}$. However, Figure 15 and §J reveal that as the number of training epochs increases, the training accuracy of SYM, RW, and ATT reach the accuracy of $\text{MAJ}_{\text{WL}}$. Because the accuracy of $\text{MAJ}_{\text{WL}}$ is often close to 100%, our experiments suggest that insufficient expressive power likely does not contribute to degree bias, drawing doubt to hypothesis **(H7)**.

Empirically inspecting the model gradients, compared to ATT, as the number of training epochs increases, the mean absolute gradients of SYM and RW are comparably small but often decrease more slowly or fluctuate. To understand this, we can analytically inspect the gradients of $\overline{\text{RW}}$:

$$\frac{\partial \ell[t]}{\partial \boldsymbol{W}^{(l)}[t]}(B[t]) = \boldsymbol{X}^T \left(\boldsymbol{P}_{\text{rw}}^l[t]\right)^T \epsilon[t]. \tag{67}$$

$\left(\boldsymbol{P}_{\text{rw}}^l[t]\right)^T$ (for $l > 0$) often has numerous eigenvalues around 0, which can yield gradients $\frac{\partial \ell[t]}{\partial \boldsymbol{W}^{(l)}[t]}(B[t])$ with a small magnitude even when $\|\epsilon[t]\|$ is not small. The same analysis holds for $\overline{\text{SYM}}$. As such, SYM and RW may get trapped in suboptimal minima during training, yielding slower or unstable convergence; in contrast, because ATT has a dynamic filter, its training loss rarely exhibits slow or unstable convergence.

## J    Additional Training Loss Plots

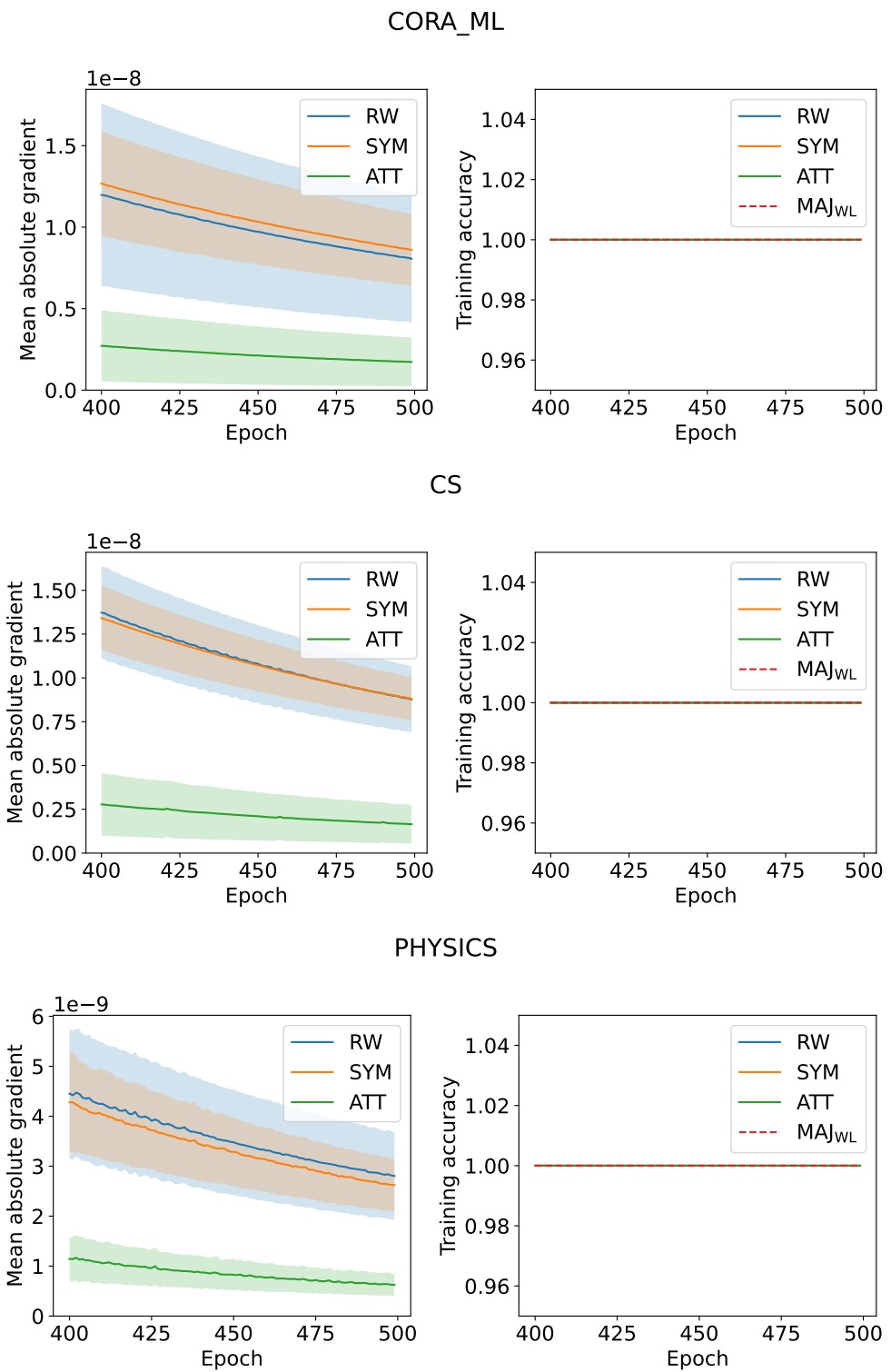

Figure 16: Mean absolute parameter gradient vs. training epoch for RW, SYM, and ATT GNNs on Cora_ML, CS, and Physics. The training accuracy of SYM, RW, and ATT ultimately reach the accuracy of MAJ$_{WL}$.

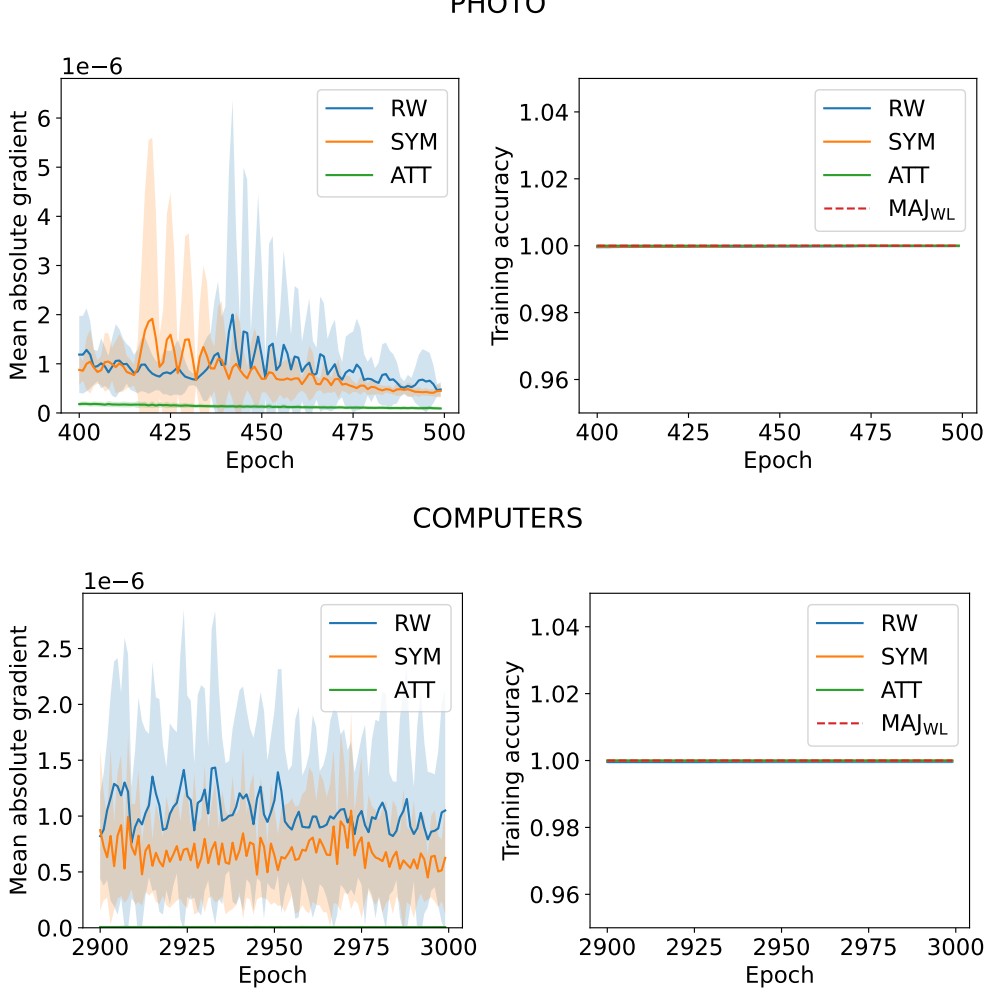

Figure 17: Mean absolute parameter gradient vs. training epoch for RW, SYM, and ATT GNNs on Photo and Computers. The training accuracy of SYM, RW, and ATT ultimately reach the accuracy of $\text{MAJ}_{\text{WL}}$.

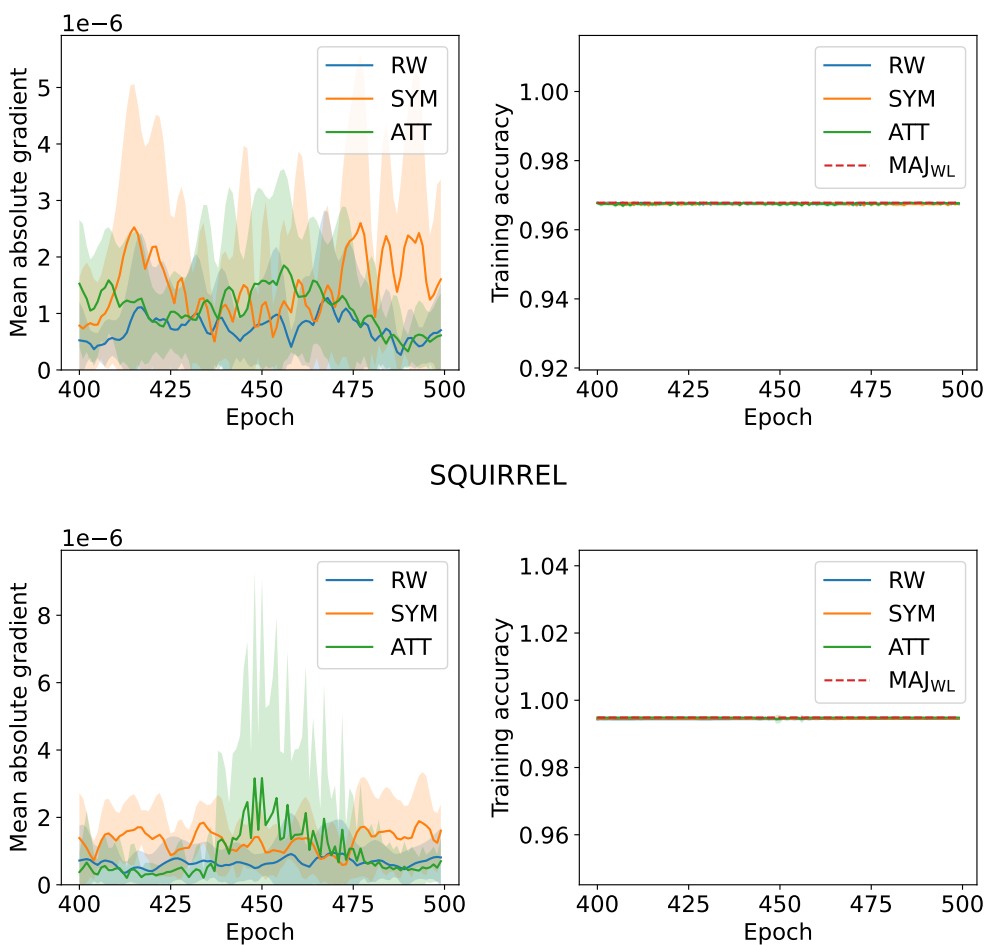

Figure 18: Mean absolute parameter gradient vs. training epoch for RW, SYM, and ATT GNNs on chameleon and squirrel. The training accuracy of SYM, RW, and ATT ultimately reach the accuracy of $MAJ_{WL}$.

## K  Connecting Inverse Collision Probability to Node Degree

Our theoretical analysis may be improved upon by establishing a rigorous connection, or a lack thereof, between the inverse collision probability of a node and its degree.

Via some preliminary analysis, we find that it is possible to express the inverse collision probability of $i$ (i.e., equivalently express the sum of $l$-hop collision probabilities) in terms of $D_{ii}$. In particular, we can show: $\sum_{l=0}^{L} \alpha_i^{(l)} = \alpha_i^{(0)} + \sum_{l=1}^{L} \sum_{j \in \mathcal{V}} [(P_{rw}^l)_{ij}]^2 = 1 + \frac{1}{D_{ii}^2} \sum_{l=1}^{L} \sum_{j \in \mathcal{V}} [\sum_{k \in \mathcal{N}(i)} (P_{rw}^{l-1})_{kj}]^2$. As before, we can see that the inverse collision probability is larger (and thus $R_{i,c'}$ is larger) when $D_{ii}$ is larger *and* $\sum_{l=1}^{L} \sum_{j \in \mathcal{V}} [\sum_{k \in \mathcal{N}(i)} (P_{rw}^{l-1})_{kj}]^2 \in o(D_{ii}^2)$. However, because $\sum_{k \in \mathcal{N}(i)}$ depends on $D_{ii}$, this expression does not completely isolate the impact of $D_{ii}$ on the inverse collision probability. A similar expression can be derived for SYM by expressing: $P_{sym} = D^{\frac{1}{2}} P_{rw} D^{-\frac{1}{2}}$.

Alternatively, one may consider $\sum_{l=0}^{L} \alpha_i^{(l)} \leq \sum_{l=0}^{\infty} \alpha_i^{(l)}$, and then plug in the steady-state probabilities of uniform random walks on graphs. However, as $l \to \infty$, $(P_{rw}^l)_{ij} = \frac{D_{jj}}{2|\mathcal{E}|}$ only depends on $D_{jj}$ (not $D_{ii}$, as desired). It is also possible to upper bound the inverse collision probability as: $1/\sum_{l=0}^{L} \alpha_i^{(l)} \leq 1/(\alpha_i^{(0)} + \alpha_i^{(1)}) = 1/(1/D_{ii} + 1)$, which is in terms of $D_{ii}$; however, we cannot use this upper bound to lower bound $R_{i,c'}$. The collision probabilities themselves are intimately related to more global properties of graphs; for instance, via eigendecomposition, $[(P_{rw}^l)_{ij}]^2 \leq \lambda^{2l}$, where $\lambda < 1$ is the eigenvalue of $P_{rw}$ with the largest magnitude [36]. Then, the inverse collision probability is strictly greater than $\frac{1}{\sum_{l=0}^{L} |\mathcal{V}| \lambda^{2l}}$.

This being said, our paper argues that the inverse collision probability is a more fundamental quantity that influences what the fair graph learning community has termed "degree bias" than degree alone, which we have shown is positively associated with the inverse collision probability.

# L   Limitations and Future Directions

**Survey**   While we aimed to be thorough in our survey of prior papers on degree bias, it is inevitable that we missed some relevant work. In addition, we extract hypotheses for the origins of degree bias from the main bodies of papers; it is possible that the hypotheses do not fully or accurately reflect the *current* perspectives of the papers' authors.

**Theoretical analysis**   Our theoretical analysis is limited to linearized message-passing GNNs. While this is a common practice in the literature [51, 7, 39], it is a strong simplifying assumption. We empirically validate our theoretical findings on GNNs with non-linear activation functions, but our paper does not address possible sources of degree bias related to these non-linearities, which would be interesting to investigate in future work. Towards this, a possible option is to assume that node features are drawn from a Gaussian distribution and derive precise high-dimensional asymptotics for degree bias in *non-linear* GNNs using the Gaussian equivalence theorem, as in [1]. Our assumptions that GNNs generalize in expectation (Theorem 1) and the variance of node representations is finite (Theorems 2 and 3) are not overly strong assumptions in practice.

Furthermore, our paper focuses on node classification. However, our findings readily lend insight into the origins of degree bias in link prediction. For example, if one uses node representations and an inner-product decoder to predict links between nodes, our results indicate that:

- In the random walk filter case, link prediction scores between low-degree nodes will suffer from higher variance because low-degree node representations have higher variance (Theorem 2). Hence, Theorem 1 suggests that predictions for links between low-degree nodes will have a higher misclassification error.

- In the symmetric filter case, our proof of Theorem 3 suggests that the link prediction scores between high-degree nodes will be over-calibrated (i.e., disproportionately large) because high-degree node representations have a larger magnitude (i.e., approximately proportional to the square root of their degree). Hence, over-optimistic and possibly inaccurate links will be predicted between high-degree nodes.

The labels and evaluation for link prediction can confound intuition. Unlike node classification, the labels for link prediction (i.e., the existence or not of a link) make the task naturally imbalanced with respect to node degree; high-degree nodes have a much higher rate of positive links than low-degree nodes. This association between degree and positive labels can influence the misclassification error. Ultimately, more rigorous theoretical analysis and experimentation are needed to confirm the hypothesized implications of node degree for link prediction performance. Similarly, more research is required to understand the implications of our findings for degree bias in the context of graph classification.

Furthermore, our theoretical analysis does not encompass heterogeneous graphs. In our literature survey, we cover works that establish the issue of degree bias for knowledge graph predictions and embeddings (e.g., [5, 43]). Our theoretical analysis is general and covers diverse message-passing GNNs, and can be extended to heterogeneous networks if messages aggregated from different edge types are subsequently linearly combined. In this setting, $R_{i,c'}$ can be computed as the sum of the prediction homogeneity quantities $(\sum_{l=0}^{L} \beta_{i,c'}^{(l)})^2$ for each edge type divided by the sum of the collision probability quantities $\sum_{j \in \mathcal{V}} [(\boldsymbol{P}_{rw}^l)_{ij}]^2$ for each edge type.

**Empirical validation**   We sought to be transparent throughout our paper regarding misalignments between empirical and theoretical findings. Our experiments focus on the transductive learning setting; it would be valuable to validate our theoretical findings in the inductive learning setting as well. Furthermore, while we aimed to cover diverse domains (e.g., citation networks, collaboration networks, online product networks, Wikipedia networks), as well as homophilic and heterophilic networks, it remains to identify the shortcomings of our theoretical findings for heterogeneous and directed networks.

**Principled roadmap**   To do justice to studying the origins of degree bias in GNNs, which our paper reveals has various and sometimes conflicting understandings, we limit the scope of our paper to understanding the root causes of degree bias, not providing a concrete algorithm to alleviate it.

Instead, we offer a principled roadmap based on our theoretical findings to address degree bias in the future. We further comment on the limitations of algorithmic solutions to degree bias in our Broader Impacts section below.

# M   Broader Impacts

Our paper touches upon issues of discrimination, bias, and fairness, with the goal of advancing justice in graph learning. In particular, our analysis of the origins of degree bias in GNNs seeks to inform principled approaches to mitigate unfair performance disparities faced by low-degree nodes in networks (e.g., lowly-cited authors, junior researchers, niche product and content creators). Despite our focus on fairness, our work can still have negative societal impacts in malicious contexts. For example, alleviating the degree bias of GNNs that are intended to surveil individuals can further violate the privacy of low-degree individuals. Ultimately, performance disparities should only be mitigated when the task is aligned with the interests and well-being of marginalized individuals; we explicitly do not support evaluating or mitigating degree bias to ethics-wash inherently harmful applications of graph learning. Furthermore, any algorithm proposed to alleviate degree bias will not be a 'silver bullet' solution; graph learning practitioners must adopt a sociotechnical approach: (1) critically examining the societal factors that contribute to their networks have degree disparities to begin with, and (2) monitoring their GNNs in deployment and continually adapting their degree bias evaluations and algorithms. In addition, alleviating degree bias does not necessarily address other forms of unfairness in graph learning, e.g., equal opportunity with respect to protected social groups [2], dyadic fairness [25], preferential attachment bias [44]; fairness algorithms are contextual and not one-size-fits-all.

