# OpenReview forum: "Theoretical and Empirical Insights into the Origins of Degree Bias in Graph Neural Networks"
_NeurIPS.cc/2024/Conference — NeurIPS 2024 poster_

### Official Review · Reviewer_Vv7z · 2024-07-07

**Soundness:** 3
**Presentation:** 4
**Contribution:** 4
**Rating:** 8
**Confidence:** 4

**Summary:**

The authors consider the problem of degree bias for node classification using graph neural networks (GNNs). Much prior work has shown that GNNs tend to be much more accurate for nodes with higher degree than for lower degree. The authors survey 38 papers that pose hypotheses, sometimes contradictory, for why this degree bias exists. In this paper, the authors provide theoretical analyses on linearized versions of message-passing GNNs using random walk (RW) and symmetric (SYM) graph filters. Their bounds yield some insights on the origins of degree bias at both test time and training time and provide evidence to support some hypotheses from prior work but not others. Finally, they run experiments using the RW, SYM, and graph attention (ATT) filters on a variety of data sets, confirming their theoretical results. They conclude by providing some criteria that mitigation strategies for degree bias should target.

**Strengths:**

- Provides a very general theoretical analysis on the misclassification probabilities of nodes using message-passing GNNs with RW and SYM graph filters.
- Empirical results support predictions from theoretical results.
- Theoretical and empirical results yield insights as to what types of mitigation strategies for degree bias may be most effective.
- Thoroughly explores hypotheses for degree bias in the existing literature and connects the theoretical results in this paper to provide evidence for some hypotheses and against others.
- Excellent use of hyperlinks to guide reader back and forth between portions of the supplement, figures, and body text. I really enjoyed reading this paper and would likely have found it much more frustrating if I had to navigate back and forth manually.
- Addresses a topic of great interest to the NeurIPS community, as illustrated by the large body of recent work on the topic.

**Weaknesses:**

- Theorem 2 bounds the squared inverse coefficient of variation $R_{i,c^′}$ based on the inverse collision probability $1/\sum_{l=0} \alpha_i^{(l)}$. There is no relationship or bound on the inverse collision probability as a function of node degree beyond a single layer, with the authors providing only empirical results. Thus, the theoretical results are not "end to end", with this one link between misclassification probability and node degree missing.
- Analysis is for linearized versions of GNNs and not the GNNs themselves.

Minor presentation issues:
- Error bars in Figure 1 and similar figures in Appendix E are not explained in the figure caption. I see that they are explained later in the beginning of Section 3. I suggest adding (or moving) the description to the figure caption.
- Figure 2: Caption lists FAIR ATT, but those plots are not shown.

**Questions:**

1. Of the four target criteria you present in Section 6, is there any ordering or predicted importance that you would recommend future work to target?
2. I noted the lack of a relationship between inverse collision probability and node degree as a weakness. Do you have any potential leads on whether it would be possible to bound this based on other quantities in the graph? This would allow your theoretical results to potentially be "end to end".

**Limitations:**

Limitations are discussed in the supplementary material. The paper would be strengthened if a short summary of the main limitations (maybe 2 or 3 lines) could be added to the conclusion.

---

> ### Author Rebuttal · Authors · 2024-08-06
>
> Thanks for your insightful comments and positive reception! Regarding the weaknesses:
>
> - This is a great and interesting question! We agree that it would be beneficial to close this missing link. Via some preliminary analysis, we find that it is possible to express the inverse collision probability (i.e., express the sum of $l$-hop collision probabilities) in terms of $D_{i i}$. In particular, we can show: $\sum_{l = 0}^L \alpha_i^{(l)} = \alpha_i^{(0)} + \sum_{l = 1}^L \sum_{j \in {\cal V}} [(P^l_{rw})\_{i j}]^2 = 1 + \frac{1}{D^2\_{i i}} \sum_{l = 1}^L \sum_{j \in {\cal V}} [\sum_{k \in {\cal N(i)}} (P^{l - 1}\_{rw})\_{k j}]^2$. As before, we can see that the inverse collision probability is larger (and thus $R_{i, c’}$ is larger) when $D_{i i}$ is larger, and $\sum_{l = 1}^L \sum_{j \in {\cal V}} [\sum_{k \in {\cal N(i)}} (P^{l - 1}\_{rw})\_{k j}]^2 \in o(D_{i i}^2)$. However, because $\sum_{k \in {\cal N(i)}}$ depends on $D_{i i}$, this expression does not completely isolate the impact of $D_{i i}$ on the inverse collision probability. A similar expression can be derived for SYM by expressing: $P_{sym} = D^{\frac{1}{2}} P_{rw} D^{-\frac{1}{2}}$.
>
> Another proof route that was considered was via the following bound: $\sum_{l = 0}^L \alpha_i^{(l)} \leq \sum_{l = 0}^\infty \alpha_i^{(l)}$, as then we can plug in the steady-state probabilities of uniform random walks on graphs. However, as $l \to \infty$, $(P^l_{rw})\_{i j} = D_{j j} / 2 |{\cal E}|$ only depends on $D_{j j}$ (not $D_{i i}$ as desired). It is also possible to upper bound the inverse collision probability as: $1 / \sum_{l = 0} \alpha_i^{(l)} \leq 1 / (\alpha_i^{(0)} + \alpha_i^{(1)}) = 1 / (1 / D_{i i} + 1)$, which is in terms of $D_{i i}$; however, we cannot use this upper bound to lower bound $R_{i, c’}$.
>
> We will continue ideating about: (1) better bounds for the inverse collision probability in terms of $D_{i i}$, or (2) an impossibility result for a bound that isolates the effect of $D_{i i}$ on $R_{i, c’}$. This being said, our paper argues that the inverse collision probability is a more fundamental quantity that influences what the community has termed “degree bias” than degree alone, which is positively associated with the inverse collision probability.
>
> - It would be interesting as future work to investigate the effects of nonlinearities on degree bias. Towards this, a possible option is to assume that node features are drawn from a Gaussian distribution and derive high-dimensional asymptotics for degree bias in _non-linear_ GNNs using the Gaussian equivalence theorem, as in [1].
>
> [1] Ben Adlam, Jeffrey Pennington Proceedings of the 37th International Conference on Machine Learning, PMLR 119:74-84, 2020.
>
> - Thanks for catching the presentation issues. We will move the explanation of the error bars to the caption of Figure 1 and the captions of the figures in Appendix E. We will likewise remove “FAIR ATT” from the caption of Figure 2.
>
> Regarding your questions:
> 1. We recommend that future work target criteria 2 and 3 (lines 291-294) first. These criteria are important because they reflect (to a large extent) inherent fairness issues with the graph filters that are popular in the graph learning community. For instance, the random walk and symmetric filters disadvantage low-degree nodes by yielding representations with high variance and low magnitude, respectively. It is valuable for the community to investigate filters that are adaptive or not restricted to the graph topology in a way that ensures that low-degree nodes are not marginalized through disparate representational distributions or poor neighborhood diversity.
>
> 2. We discuss potential leads above to bound the inverse collision probability in terms of node degree. The collision probabilities themselves are intimately related to (more global) properties of the graph; for instance, via eigendecomposition, $[(P^l_{rw})\_{i j}]^2 \leq \lambda^{2 l}$, where $\lambda < 1$ is the eigenvalue of $P_{rw}$ with the largest magnitude [2]. Then, the inverse collision probability is strictly greater than $\frac{1}{\sum_{l = 0}^L |{\cal V}| \lambda^{2 l}}$.
>
> [2] Lovasz, L. M. Random walks on graphs: A survey. 2001.
>
> 3. We will definitely detail the main limitations of our paper in the conclusion of the camera-ready version.

---

> > ### Comment · Reviewer_Vv7z · 2024-08-13
> >
> > Thank you for the detailed response. I continue to strongly support this paper and do not view the current gap on bounding the inverse collision probability as a major issue that should prevent this paper from being published. I also support the authors plans to address the weaknesses in future work. I see no reason to change my score.

---

### Official Review · Reviewer_USpG · 2024-07-12

**Soundness:** 4
**Presentation:** 3
**Contribution:** 3
**Rating:** 7
**Confidence:** 3

**Summary:**

The paper provides a comprehensive analysis of the origins of degree bias in message-passing GNNs, proving that high-degree test nodes have a lower probability of misclassification, regardless of how GNNs are trained.  They surveyed 38 papers on degree bias and found that existing hypotheses are often not rigorously validated and can be contradictory.  They also show that some GNNs may adjust their loss on low-degree nodes more slowly during training, but with sufficient training epochs, these models can achieve their maximum possible training accuracy. The theoretical findings are validated on eight real-world networks, and a roadmap to alleviate degree bias is proposed.

**Strengths:**

+This paper explores the important issue of the origins of degree bias in Graph Neural Networks (GNNs).

+The authors conduct a comprehensive survey on degree bias, carefully identifying and validating existing hypotheses while pointing out those that are contradictory or lack validation.

+The study examines the effects of different graph filters and analyzes degree bias separately during both test time and training time.

+The paper provides a thorough theoretical analysis and supports its claims with empirical studies on eight datasets.

**Weaknesses:**

-It appears that the findings in this paper are limited to homophilous graphs, as the results do not extend to heterophilic graphs, according to the paper's analysis.

-The current analysis is based on graphs with a single edge type. Can these findings be expanded to heterogeneous graphs, where the edges can have different types? It could be worth studying to explore this possibility.

**Questions:**

Please address the questions regarding weaknesses.

**Limitations:**

Yes

---

> ### Author Rebuttal · Authors · 2024-08-06
>
> Thanks for your thoughtful feedback and questions! Regarding the weaknesses and your questions:
> - Our findings do extend to heterophilic graphs. For example, in Lines 220-222, we explain that high-degree nodes in heterophilic networks do not have more negative L-hop prediction homogeneity levels due to higher local heterophily, hence we do not necessarily observe better performance for them; we empirically validate that there is a lack of degree bias for the heterophilic datasets Chameleon and Squirrel in Figure 5. Furthermore, in lines 693-696, we explain that models do not learn more rapidly for high-degree nodes under heterophily (Figures 11-12) because the “features of nodes in the neighborhoods of high-degree nodes and training nodes are dissimilar.” None of our theoretical analysis assumes homophilic networks. We will clarify any parts of the paper that suggest that our theoretical analysis does not encompass heterophilic networks.
>
> - We agree that it is valuable to study the possibility of expanding our findings to heterogeneous graphs. For example, in our literature survey, we cover works that establish the issue of degree bias for knowledge graph predictions and embeddings (e.g., [4, 38]). Furthermore, we state in Section K, “it remains to identify the shortcomings of our theoretical findings for heterogeneous and directed networks.” Our theoretical analysis is general and covers diverse message passing GNNs, and can be extended to heterogeneous networks if messages aggregated from different edge types are subsequently linearly combined. In this setting, $R_{i, c’}$ can be computed as the sum of the prediction homogeneity quantities $(\sum_{l = 0}^L \beta_{i, c’}^{(l)})^2$ for each edge type divided by the sum of the collision probability quantities $\sum_{j \in {\cal V}} [(P_{rw}^l)_{i j}]^2$ for each edge type.
>
> [4] Stephen Bonner, Ufuk Kirik, Ola Engkvist, Jian Tang, and Ian P Barrett. Implications of topological imbalance for representation learning on biomedical knowledge graphs. Briefings in Bioinformatics, 23(5):bbac279, 07 2022.
>
> [38] Harry Shomer, Wei Jin, Wentao Wang, and Jiliang Tang. Toward degree bias in embedding based knowledge graph completion. In Proceedings of the ACM Web Conference 2023, WWW ’23, page 705–715, New York, NY, USA, 2023. Association for Computing Machinery.

---

> > ### Comment · Reviewer_USpG · 2024-08-08
> >
> > Thanks for your response. I will maintain my score.

---

### Official Review · Reviewer_Xp6E · 2024-07-12

**Soundness:** 3
**Presentation:** 3
**Contribution:** 2
**Rating:** 6
**Confidence:** 4

**Summary:**

The paper explores the causes and characteristics of degree bias in graph neural networks (GNNs), particularly in node classification tasks. It contributes to the field by providing a rigorous theoretical analysis supported by empirical evidence across multiple datasets, revealing that degree bias is influenced by factors such as homophily and diversity of neighbors. The study also proposes a structured approach to mitigate this bias, enhancing the fairness and effectiveness of GNN applications in social and recommendation systems.

**Strengths:**

originality: medium
quality: good
clarity: good
significance: medium

**Weaknesses:**

Some degree-related papers should be included and discussed in this paper, for example:

[1] takes a unified view to explain the over-smoothing and heterophily problems simultaneously by profiling nodes with two metrics: the relative degree of a node (compared to its neighbors) and the node-level heterophily.

[2] investigates how does node degree, homophily and class variance influence the node distinguishability.

[3] finds that the effectiveness of graph convolution operations in enhancing separability is determined by the Euclidean distance of the neighborhood distributions and the square root of the average node degree. Furthermore, they find that topological noise negatively affects separability by effectively lowering the average node degree.

[4] observes that the prediction accuracy of high-degree nodes is usually significantly lower under heterophily and the hypothesis that "nodes with higher degrees are naturally favored by GNNs" no longer holds. In fact, the degree-wise bias is more sensitive, but not necessarily beneficial, to high-degree nodes with regard to varying graph conditions.



[1] Two sides of the same coin: Heterophily and oversmoothing in graph convolutional neural networks. In2022 IEEE International Conference on Data Mining (ICDM) 2022 Nov 28 (pp. 1287-1292). IEEE.

[2] When Do Graph Neural Networks Help with Node Classification? Investigating the Homophily Principle on Node Distinguishability. Advances in Neural Information Processing Systems. 2024 Feb 13;36.

[3] Understanding Heterophily for Graph Neural Networks. In Forty-first International Conference on Machine Learning.

[4] Liao N, Liu H, Zhu Z, Luo S, Lakshmanan LV. Benchmarking Spectral Graph Neural Networks: A Comprehensive Study on Effectiveness and Efficiency. arXiv preprint arXiv:2406.09675. 2024 Jun 14.

**Questions:**

1. How does homophily relate to degree bias? I expect some conclusions in this paper but cannot find them.

2. The "expected similarity of the neighborhoods" in section 5 is very similar to the similarity matrix defined in [5]

3. The criteria proposed in section 6 should be verified in real-world datasets.




[5] Revisiting heterophily for graph neural networks. Advances in neural information processing systems. 2022 Dec 6;35:1362-75.

**Limitations:**

the authors adequately addressed the limitations

---

> ### Author Rebuttal · Authors · 2024-08-06
>
> Thanks for your feedback and the helpful resources! Regarding the weaknesses, we will definitely include and discuss the recommended references on homophily in our paper:
>
> [1] provides a complementary perspective on the possible performance issues of GNNs that arise from degree disparities in graphs (e.g., low-degree nodes induce oversmoothing in homophilic graphs, while high-degree nodes induce oversmoothing in heterophilic networks). Oversmoothing is related to prediction homogeneity $(\sum_{l = 0}^L \beta_{i, c’}^{(l)})^2$ (line 149); for homophilic networks, as the number of layers in a GNN increases (i.e., as oversmoothing occurs), $(\sum_{l = 0}^L \beta_{i, c’}^{(l)})^2$ gets closer to 0 (i.e., does not increase $R_{i, c’}$), thereby not inducing as much degree bias. In contrast, our theoretical analysis demonstrates that degree bias occurs without oversmoothing and is amplified by high local homophily (lines 170-171).
>
> [2] connects node distinguishability to node degree and homophily by analyzing the intra-class vs. inter-class embedding distance. We discuss similar quantities in lines 175-189 and lines 208-226, and will integrate connections to [2] in these discussions. However, with the exception of Section 3.5, [2] considers the CSBM-H model in its theoretical analysis, which has pitfalls (as we discuss in lines 79-83). Moreover, unlike our work, [2] does not explicitly link the misclassification error of a node to its degree in a more general data and model setting.
>
> [3] analyzes the effect of heterophily on GNNs via class separability, which it characterizes through neighborhood distributions and average node degree. Like with [1] and [2], we will discuss connections between prediction homogeneity, homophily, and separability. Notably, similar to [2], [3] only considers the HSBM model in its theoretical analysis.
>
> [4] observes that GNN performance is lower for high-degree nodes under heterophily. We likewise observe this in Figure 5 for Chameleon and Squirrel (which are heterophilic networks), and we will connect this observation to [4]. Moreover, our theoretical analysis explains why degree bias is not observed for heterophilic graphs; in lines 220-222, we explain that high-degree nodes in heterophilic networks do not have lower $l$-hop prediction homogeneity levels due to higher local heterophily, hence we do not necessarily observe better performance for them compared to low-degree nodes. We would additionally like to note that [4] was released on 4 Jun 2024, which was after the NeurIPS submission deadline on 22 May 2024.
>
> Regarding your questions:
>
> 1. We make connections between homophily and degree bias in the paper. For example:
> - Lines 170-171: We show that to amplify degree bias, we need to make $R_{i, c’}$ larger, for which it is sufficient to make the prediction homogeneity $\sum_{l = 0}^L \beta_{i, c’}^{(l)}$ more negative, “e.g., when most nodes in the $l$-hop neighborhood of $i$ are predicted to belong to class $c$.” This can occur when node $i$ has high local homophily.
> - Lines 175-179: Our proof of Theorem 2 shows that, in expectation, $\overline{RW}$ “produces similar representations for low and high-degree nodes with similar L-hop neighborhood homophily levels.” However, low-degree nodes have a higher representation variance (due to a lower inverse collision probability), which can amplify degree bias. This entails that factors beyond homophily (e.g., diversity of neighbors) induce degree bias.
> - Lines 208-214, 222-226: Our proof of Theorem 3 likewise shows that for SYM, homophily alone does not induce degree bias.
> -  Lines 220-222: We explain that high-degree nodes in heterophilic networks do not have lower $l$-hop prediction homogeneity levels due to higher local heterophily, hence we do not necessarily observe better performance for them.
> - Lines 693-696: We explain that models do not learn more rapidly for high-degree nodes under heterophily (Figures 11-12) because the “features of nodes in the neighborhoods of high-degree nodes and training nodes are dissimilar.”
>
> 2. The similarity matrix defined by [5] is _post-aggregation_, while our matrix is _pre-aggregation_. Moreover, our similarity matrix is an interpretable quantity that naturally arose during our theoretical analysis. We are happy to cite [5].
>
> 3. We perform extensive validation of the criteria in Section 6 on 8 real-world datasets:
> - Figure 3 and the plots in Section G show strong positive associations between inverse collision probability and degree for the RW, SYM, and ATT filters, and Figure 1 and the plots in Section E show strong negative associations between degree and test loss for the homophilic datasets. Hence, we validate that there is a negative association between inverse collision probability and test loss.
>
> - The lack of degree bias observed in Figure 5 for Chameleon and Squirrel (which are heterophilic networks), compared to Figure 1 and the other plots in Section E, confirms our theoretical finding that under heterophily, the prediction homogeneity for high-degree nodes is closer to 0, so high-degree nodes do not necessarily experience better performance.
>
> - Figures 2, 6-10 empirically confirm our theoretical finding that disparities in the expectation and variance of node representations are responsible for performance disparities. Figures 11-12 suggest that smaller distributional differences among representations (due to heterophily) can alleviate degree bias.
>
> - Figure 2 and the plots in Section F empirically validate the training discrepancies between low and high-degree nodes that we theoretically analyze in Section 5.

---

> > ### Comment · Reviewer_Xp6E · 2024-08-13
> >
> > Thanks for  the  detailed response and I will raise my rating to 6.

---

### Official Review · Reviewer_Ao8B · 2024-07-24

**Soundness:** 3
**Presentation:** 3
**Contribution:** 3
**Rating:** 5
**Confidence:** 4

**Summary:**

Given the wide-adoption of GNN-based node classification and the potential risk of degree-related bias, this paper systematically reviews the degree-bias paper in previous literature and proposes a theoretical framework to analyze the degree-related bias. Several insightful observations have been drawn with empirical verification.

**Strengths:**

(1) A systematic investigation of a long-standing yet not addressed research problem, degree bias. Comprehensive summary of previous works on this has been presented

(2) The degree-related bias is urgently related to social benefit.

**Weaknesses:**

(1) Some experimental results are not supportive of some claims.

(2) Some theoretical analyses lack intuitive justification from a graph topology perspective, e.g., the connection with network homophily/heterophily.

**Questions:**

(1) Since the degree bias focused on here is node classification, would it be better to note this special case somewhere in the paper, or at least for the title, would it be better to reword and limit the task scope to node classification? Is there any insight or potential thought on degree-related bias in link prediction and graph classification (here the degree would correspond to graph density)?

(2) In Figure 5, when looking at the Squirrel and Chameleon datasets, there is no significant difference in high-degree and low-degree test loss, is there a specific reason for their difference from the other datasets?

(3) Maybe it is a typo, but in the Equation at the bottom of page 4, if we unify different layers' weight transformation matrices as of dimension $\mathbb{R}^{d^0\times C}$, would it cause some dimension mismatch during matrix multiplication? Why does every layer of convolution consider the project from space $\mathbb{R}^{d^0}$ to $\mathbb{R}^{C}$? Moreover, it might be better to add back the Equation number.

(4) It is not so clear about "the diversity of neighborhoods" until the formal definition in Line 168. It might be better to provide some concise illustrations to help readers understand the diversity of neighbors when it was mentioned for the very first time.

(5) Based on Theorem 2, in addition to decreasing the negative $\sum_{l=0}^{L}\beta_{i, c'}^{(l)}$ so that $R_{i, c'}$ could increase, could we also increase the positive $\sum_{l=0}^{L}\beta_{i, c'}^{(l)}$ so that $R_{i, c'}$ could increase? Are these two scenarios corresponding to some graph local structure? Based on my understanding, as $\beta_{i, c'}^{(l)}$ measures of the local subgraph difference between class $c'$ and class $c$, it really boils down to the local homophily and local heterophily, if the value is pretty small, then it might correspond to a very small $\beta$, which might correspond to a equal contribution between class c and c' and hence the subgraph is really like a mixture between nodes from these two classes and hence cannot work very well. To sum up this point, my thinking is that it would be better to provide some intuition for some theoretical findings from a topology perspective.

(6) It might be interesting to add some analysis when a distribution shift happens.

(7) For the training loss visualization in Figure 2, it is really hard to see the difference of low/high-degree nodes for RW and ATT.

**Limitations:**

The paper addressed all limitations with the following exceptions:

The paper mainly studies degree-related bias for node classification. There are some other tasks such as link prediction and graph classification worth similar analysis as well.

---

> ### Author Rebuttal · Authors · 2024-08-06
>
> Thanks for your comments and thorough questions! Regarding your questions:
>
> (1) We will make the scope of the paper clearer by indicating in the title, introduction section, and limitations section that we focus on node classification. Our findings readily lend insight into the origins of degree bias in link prediction. For example, if one uses node representations and an inner-product decoder to predict links between nodes, our results (lines 208-226) indicate that:
> - In the random walk filter case, link prediction scores between low-degree nodes will suffer from higher variance because low-degree node representations have higher variance. Hence, Theorem 1 suggests that predictions for links between low-degree nodes will have a higher misclassification error.
> - In the symmetric filter case, the link prediction scores between high-degree nodes will be over-calibrated (i.e., disproportionately large) because high-degree node representations have a larger magnitude (i.e., approximately proportional to the square root of their degree). Hence, our proof of Theorem 3 suggests that over-optimistic and possibly inaccurate links will be predicted between high-degree nodes.
> More research is required to understand the implications of our findings for degree bias in the context of graph classification; such research likely has rich connections to the literature on spectral expressive power [1].
>
> [1] Balcilar, Muhammet, et al. "Analyzing the expressive power of graph neural networks in a spectral perspective." 2021.
>
> (2) The reason that we do not observe degree bias for Chameleon and Squirrel is because unlike the other datasets, these datasets are heterophilic. We intentionally include these datasets to draw contrast to the other, homophilic datasets and validate our theory. For example, in lines 220-222, we explain that high-degree nodes in heterophilic networks do not have more negative $l$-hop prediction homogeneity levels due to higher local heterophily levels; hence, we do not necessarily observe better performance for them compared to low-degree nodes.
>
> (3) We will ensure that equation numbers are visible for all equations. The weight transformation matrices should all be of dimension $d_0 \times C$; there is not a dimension mismatch issue. This is because we consider linearized versions of message-passing GNNs. In particular, in the equation between lines 132 and 133, if we set each $\sigma^{(l)}$ to be identity and recursively expand the matrix multiplications, we end up with the expression: $Z^{(L)} = \sum_{l = 0}^L P_{rw}^l X W^{(l)}$, where $W^{(l)}$ is the sum of all the weight terms in the expansion that correspond to $P_{rw}^l$; for simplicity, we collapse each sum of weight terms into a single weight matrix. Because each $W^{(l)}$ maps the input features $P_{rw}^l X \in \mathbb{R}^{n \times d_0}$ to the outputs $Z^{(l)} \in \mathbb{R}^{n \times C}$, they must all be of size $d_0 \times C$. It makes sense to have a different weight matrix for each $P_{rw}^l X$, as we may need to extract different information from features aggregated from neighborhoods at different hops.
>
> (4) This is great feedback. We will include a figure at the beginning of the paper that visually illustrates the concepts of prediction homogeneity and collision probability, and their connections to homophily and diversity.
>
> (5) In Theorem 2 (lines 118-119), we assume that $\mathbb{E} [Z_{i, c’}^{(L)} - Z_{i, c}^{(L)}] < 0$ (i.e., the model generalizes in expectation); this is necessary to make a mathematically rigorous statement about degree bias via tail bounds. Thus, we cannot make $\sum_{l = 0}^L \beta_{i, c’}^{l)}$ more positive to increase $R_{i, c’}$, as this would violate the assumption of our theorem. However, intuitively, it also would not make sense that RW and SYM reduce the misclassification error for a node by predicting its neighbors to be of a different class, since message passing smooths the representations of adjacent nodes.
>
> Furthermore, per lines 153-154, “$\beta_{i, c’}^{(l)}$ measures the expected prediction score for nodes j, weighted by their probability of being reached by a _length-l random walk_ starting from i.” Because $\beta_{i, c’}^{(l)}$ depends on the distribution of random walks from $i$, it is intimately related to local graph structure. Indeed, $\beta_{i, c’}^{(l)}$ can be interpreted as a “local subgraph difference” and is highly influenced by the local homophily of $i$ (as we say in lines 170-174). However, $\beta_{i, c’}^{(l)}$ is also influenced by the presence of $l$-hop neighbors contained in the training set, as the model is more likely to correctly classify these nodes by a large margin (lines 172-173); hence, $\beta_{i, c’}^{(l)}$ does not _only_ boil down to local homophily. When revising our paper, we will include and expand on these clarifications on the similarities and differences between $\beta_{i, c’}^{(l)}$ and local homophily, to provide better intuition from a topological perspective.
>
> (6) We can add some analysis for distribution shifts. For example, our results in Section 4 can be built upon to show that shifts in local homophily from train to test time reduce test-time prediction performance, which can bring $\beta_{i, c’}^{(l)}$ closer to 0; this can increase $R_{i, c’}$, thereby not inducing as much degree bias.
>
> (7)  We will make the training loss visualizations in Figure 2 larger. The primary takeaway from this figure is that, in the case of RW and ATT, the training loss curves for low and high-degree nodes (including error bars) overlap during the first ~20 epochs of training; however, for SYM, the loss curve for high-degree nodes descends more rapidly than the curve for low-degree nodes.
>
> Regarding the weaknesses:
>
> (1) Could you please elaborate on which experimental results are “not supportive of claims?” We would like to address or clarify any potential mismatches between our theoretical analysis and experiments.
>
> (2) Please see (5) above.

---

> > ### Comment · Reviewer_Ao8B · 2024-08-11
> > **Thank you for your response!**
> >
> > Despite most of the concerns have been addressed, I still have follow-up questions:
> >
> > (1) **In the random walk filter case, link prediction scores between low-degree nodes will suffer from higher variance because low-degree node representations have higher variance. Hence, Theorem 1 suggests that predictions for links between low-degree nodes will have a higher misclassification error.**, there is some other work proving that local clustering coefficient of nodes decreases as node degree increases and since LCC is very related to node link prediction performance, I am not sure whether this intuition is right or not.
> >
> > (2) **Could you please elaborate on which experimental results are “not supportive of claims?” We would like to address or clarify any potential mismatches between our theoretical analysis and experiments.** In Figure 6 middle row, I didn't see the training loss decrease more rapidly for SYM on low-degree nodes as claimed in the main paper.

---

> > > ### Author Response · Authors · 2024-08-13
> > > **Thank you for your follow-up questions!**
> > >
> > > We are glad that most of your concerns have been addressed! Regarding your follow-up questions:
> > >
> > > (1) Thanks for bringing up this interesting perspective. Indeed, it has been observed that the local clustering coefficient (LCC) decreases as node degree increases [1]. This is in part due to real-world networks being sparse and high-degree nodes inducing a larger number of possible connections between neighbors (i.e., a larger denominator when computing the LCC).
> > >
> > > Some works have studied the impact of clustering coefficients on link prediction performance [2, 3, 4]. However, these papers only consider the effect of the *global* clustering coefficient of a network on overall link prediction performance for that network. That is, these papers find that the overall link prediction performance of network embedding algorithms is often higher for networks with a larger global clustering coefficient (and even then, this is not observed for some algorithms like Matrix Factorization [2]). These papers do not consider disparities in link prediction performance across nodes in the same network with different local clustering coefficients; the trend of better link prediction performance with a higher clustering coefficient may not hold locally because the global clustering coefficient is a simple average of and does not account for variance in local clustering coefficients across nodes. Furthermore, [2, 3, 4] do not consider initial node features or graph neural networks, which often have a narrower receptive field than spectral embedding methods (e.g., random walk, eigendecomposition).
> > >
> > > Moreover, the labels and evaluation for link prediction can confound intuition. Unlike node classification, the labels for link prediction (i.e., the existence or not of a link) make the task naturally imbalanced with respect to node degree; high-degree nodes have a much higher rate of positive links than low-degree nodes. This association between degree and positive labels can influence the misclassification error. In addition, many published link prediction evaluation results are based on label sampling methods that favor high-degree nodes [5].
> > >
> > > Ultimately, more rigorous theoretical analysis and experimentation are needed to confirm the implications of node degree for link prediction performance.
> > >
> > > (2) The training loss curves in Figure 6 still support our theoretical analysis. Theorem 4 reveals that for $\overline{SYM}$, node degree _and_ the (degree-discounted) expected feature similarity $\tilde{\chi}_i$ affects the rate of learning. On the other hand, Theorem 5 indicates that for $\overline{RW}$, while we do not expect node degree to impact the rate of learning, the expected feature similarity $\chi_i$ is still influential. Hence, interpreting Theorems 4 and 5 jointly, we expect and accordingly observe that the orange curve for SYM has a steeper rate of decrease *relative* to the orange curve for RW as the number of epochs increases. We will revise the interpretation of our results to make it more clear that while node degree highly affects the rate of learning, differences in $\chi_i$ across nodes of different degrees are also influential (lines 268-273).
> > >
> > > [1] Vázquez, Alexei, Romualdo Pastor-Satorras, and Alessandro Vespignani. "Large-scale topological and dynamical properties of the Internet." Physical Review E 65.6 (2002): 066130.
> > >
> > > [2] Robledo, O.F., Zhan, XX., Hanjalic, A. et al. Influence of clustering coefficient on network embedding in link prediction. Appl Netw Sci 7, 35 (2022). https://doi.org/10.1007/s41109-022-00471-1
> > >
> > > [3] Feng, Xu, J. C. Zhao, and Ke Xu. "Link prediction in complex networks: a clustering perspective." The European Physical Journal B 85 (2012): 1-9.
> > >
> > > [4] M. Khosla, V. Setty and A. Anand, "A Comparative Study for Unsupervised Network Representation Learning," in IEEE Transactions on Knowledge and Data Engineering, vol. 33, no. 5, pp. 1807-1818, 1 May 2021, doi: 10.1109/TKDE.2019.2951398.
> > >
> > > [5] Aiyappa, Rachith, et al. "Implicit degree bias in the link prediction task." arXiv preprint arXiv:2405.14985 (2024).

---

### Author Rebuttal · Authors · 2024-08-06

We thank the reviewers for their thoughtful and helpful comments! We are pleased to read that:
- Reviewer Ao8B finds our paper constitutes a “systematic investigation of a long-standing yet not addressed research problem.”
- Reviewer Xp6E finds our paper has good quality and clarity.
- Reviewer USpG finds our paper “provides a thorough theoretical analysis and supports its claims with empirical studies on eight datasets.”
- Reviewer Vv7z finds our paper “provides a very general theoretical analysis on the misclassification probabilities of nodes” and “really enjoyed reading this paper.”

We address the weaknesses and questions raised by each reviewer in individual responses below.

---

### Decision · Program_Chairs · 2024-09-25

**Decision:**

Accept (poster)

**Comment:**

This paper provides theoretical results to justify why Graph Neural Networks perform better for high-degree nodes. The authors place their theoretical findings in the context of previously proposed hypothesis for the origins of degree bias. They validate their findings on real-world networks and give a sketch of how to tackle degree bias. The reviewers find the studied problem to be an important and timely one. The authors should incorporate the suggestions made by the reviewers to strengthen the paper.